# Generative Diffusion Prior Distillation for Long-Context Knowledge Transfer

**Nilushika Udayangani**[*]**, Kishor Nandakishor & Marimuthu Palaniswami**
Department of Electrical and Electronic Engineering
University of Melbourne
Parkville, VIC 3052, Australia

## Abstract

While traditional time-series classifiers assume full sequences at inference, practical constraints (latency and cost) often limit inputs to partial prefixes. The absence of class-discriminative patterns in partial data can significantly hinder a classifier's ability to generalize. This work uses knowledge distillation (KD) to equip partial time series classifiers with the generalization ability of their full-sequence counterparts. In KD, high-capacity teacher transfers supervision to help student learning on the target task. When the generalization gap is due to limited parameter capacity, matching with teacher features has shown promise. However, when the generalization gap stems from training-data differences (full versus partial), the teacher's *full-context features* can be an overwhelming target signal for the student's *short-context features*. To provide progressive, diverse, and collective teacher supervision, we propose **G**enerative **D**iffusion **P**rior **D**istillation (**GDPD**), a novel KD framework that treats short-context student features as degraded observations of the target full-context features. Inspired by the iterative restoration capability of diffusion models, we learn a diffusion-based generative prior over teacher features. Leveraging this prior, we (posterior-)sample target teacher representations that could best explain the missing long-range information in the student features and optimize the student features to be minimally degraded relative to these targets. GDPD provides each student feature with a distribution of task-relevant long-context knowledge, which benefits learning on the partial classification task. Extensive experiments across earliness settings, datasets, and architectures demonstrate GDPD's effectiveness for full-to-partial distillation.

## 1 Introduction

Many real-world applications in healthcare and industrial automation rely on supervised classification of time series, where the goal is to accurately assign a class label to a given sequence. While traditional models assume access to the entire sequence during inference, this assumption often breaks down in practical settings. In many scenarios, models see only a prefix of the time series due to constraints such as latency, cost, or sensor dropout. For instance, in emergency arrhythmia detection from ECG, decisions may need to be made from 5–10 seconds of data rather than a full 60-second recording. However class-discriminative patterns may emerge at any point in the sequence, and missing these patterns under partial observability reduces class separability, causing classifiers trained and operated on partial data to generalize poorly. This work investigates how supervised classifiers, trained and operated on partial time series can be effectively equipped with the capacity to generalize from full-length series.

For a partial (prefix) time series, the true class is ambiguous since multiple classes can appear identical in the early timesteps before diverging later. Therefore, when training a classifier on partial data, hard-label supervision alone can be misleading, causing the model to overfit to spurious early cues and form unstable decision boundaries. To prevent this, we propose to provide additional regularization signal from a teacher model trained on full-length sequences, inspired by Knowledge

---

[*]Corresponding Author. Email: hewadehigaha@student.unimelb.edu.au

Distillation (KD). KD, first introduced in Buciluǎ et al. (2006); Hinton et al. (2015), is a training paradigm in which knowledge is transferred from a teacher to guide the training of a student network. The teacher is a model that learns representations which generalize well—an ability acquired through computationally intensive training or greater architectural capacity. This ability can be distilled into a student, which may lack the inductive biases to discover such representations from training data alone, often due to limited training resources or parameter capacity. The most widely adopted approach to KD trains the student to match the teacher's output logits, providing an additional regularization signal during optimization on the target task (Hinton et al., 2015). Later works (Romero et al., 2015; Park et al., 2019; Zagoruyko & Komodakis, 2016) extended this idea to match intermediate features beyond the output logits.

However, these direct feature/logit matching KD methods were proposed to address the generalization gap arising from differences in parameter capacity, with both the student and the teacher privileged to see the same data. In contrast, when distilling knowledge from a teacher trained on full-length sequences to a student trained on partial sequences, several fundamental concerns arise: **1) Can the distillation technique transfer the teacher knowledge effectively?** Even when the teacher is a highly capable model with strong representational quality, the student may fail to properly comprehend the transferred knowledge, leading to limited gains from KD (Cho & Hariharan, 2019; Mirzadeh et al., 2020; Qiu et al., 2022; Stanton et al., 2021). Prior research has attributed this issue to the capacity/architectural gap between teacher and student, and proposed intermediate "teacher-assistant" models (Mirzadeh et al., 2020; Son et al., 2021) and student-friendly teacher training (Park et al., 2021; Rao et al., 2023; Cho & Hariharan, 2019). However, when the teacher and student are exposed to different input spaces (e.g., full versus partial data), an inherent representational gap is introduced even when the models have identical parameter capacity. In such scenarios, if the distillation loss is poorly designed by directly enforcing alignment with the teacher's full-context features, it can overwhelm the student, which only encodes partial-context features, and thereby limit its ability to effectively absorb the transferred knowledge.

**2) Is a single teacher's perspective diverse enough?** Exposing students to diverse yet consistent views of the same underlying information enhances generalization and fosters robust inductive biases (You et al., 2017; Allen-Zhu & Li, 2020; Hossain et al., 2025). While existing works promote diversity through teacher ensembles (Allen-Zhu & Li, 2020; You et al., 2017) or mutual supervision among student ensembles (Zhang et al., 2018; Furlanello et al., 2018), a key concern remains whether supervision from a single model provides sufficient diversity of knowledge. Hossain et al. (2025) demonstrate that multiple augmented teacher views can be generated from a single model by perturbing its features with random noise. This increases knowledge diversity while avoiding the cost of retraining multiple models. Providing diverse perspectives of teacher knowledge is particularly important in full-to-partial distillation, where student features are degraded, incomplete, or ambiguous compared to those of the teacher, as we do not want to overcommit to a single possible interpretation of the missing or noisy information.

**3) Is the knowledge faithful?** KD transfers limited knowledge leading students with very different predictive distributions from their teachers, hindering safe substitution (Stanton et al., 2021; Lamb et al., 2023; Hewa Dehigahawattage et al., 2025). Even though knowledge improves predictive accuracy, achieving good *fidelity*, the ability of the student to match teacher predictions, with existing methods is extremely difficult (Stanton et al., 2021). Stanton et al. (2021) observe that augmenting the distillation set with data samples not present in the teacher's training data increases the drop in distillation fidelity. Similarly, the training-data mismatch that arises when the teacher is exposed to full-length data while the student is exposed to partial data can make it more challenging for the student to match the teacher's predictive distribution.

With these concerns in mind, we rethink distillation from a different lens and propose **Generative Diffusion Prior Distillation (GDPD)**. In GDPD, we view student representations learned from partial sequences as *degradations* (partial measurements) of target teacher features derived from full-length sequences. Inspired by the iterative restoration power of diffusion models (Kawar et al., 2022), we train a diffusion model to serve as a generative prior over teacher features, capturing and storing their statistical structure. Using this prior, we search within the space of teacher features for target representations with optimal teacher knowledge, and train student features to become minimally degraded relative to these discovered targets. Unlike conventional KD, which provides a single teacher signal, we model knowledge as a distribution over target teacher signals. We discuss how this distributional knowledge helps GDPD to address above three concerns of KD, exacerbated

in full-to-partial distillation, by generating teacher signals that 1) are dynamic and progressive with respect to the student's current capability, 2) provide stochastic diversity of the same features, and 3) complete optimal knowledge through collective aggregation (Section 3.4).

In short, we make following **contributions**: 1) demonstrating that KD can equip early time-series classifiers, operating on partial time series, with the generalization ability of classifiers trained on full-length time series, establishing this direction as a pioneer effort, 2) being the first to model teacher knowledge as a generative distribution, formulating the target teacher–student feature relationship as an ill-posed problem 3) introducing a novel KD framework, GDPD, to provide dynamic, diverse, and collective knowledge for effective full-to-partial distillation, and 4) providing an in-depth analysis and discussion evaluating GDPD and baseline KD methods in full-to-partial distillation.

## 2 PRELIMINARY

**Knowledge Distillation.** KD seeks optimal student parameters by jointly minimizing the task loss $\mathcal{L}_{\text{Task}}$ and a distillation loss $\mathcal{L}_{\text{KD}}$ that aligns the student with a pre-trained teacher:

$$\boldsymbol{\theta}_* = \arg\min_{\boldsymbol{\theta}} \ \lambda_{\text{Task}} \, \mathcal{L}_{\text{Task}}(\boldsymbol{\theta}) + \lambda_{\text{KD}} \, \mathcal{L}_{\text{KD}}(\boldsymbol{\theta}), \tag{1}$$

where $\lambda_{\text{Task}}$ and $\lambda_{\text{KD}}$ control the relative contributions of the two terms.

**Diffusion Models.** Given samples from the data distribution $p_{\text{data}}$, diffusion models are capable of learning a parameterized distribution $p_\phi$ that approximates $p_{\text{data}}$ and is easy to sample from (Song et al., 2020b). This is achieved through forward diffusion and reverse denoising processes. *The Forward Process* is a Markov chain that gradually corrupts data $\mathbf{z}_0 \sim p_{\text{data}}$ until it approaches Gaussian noise $\mathbf{z}_T \sim p_{\text{latent}} = \mathcal{N}(0, \mathbf{I})$ after $T$ diffusion steps. Corrupted latent variables $\mathbf{z}_1, \cdots, \mathbf{z}_T$ are sampled from $p_{\text{data}}$ with a diffusion process defined as a chain of Gaussian transitions:

$$q(\mathbf{z}_{1:T} \mid \mathbf{z}_0) = \prod_{t=1}^{T} q(\mathbf{z}_t \mid \mathbf{z}_{t-1}), \qquad q(\mathbf{z}_t \mid \mathbf{z}_{t-1}) = \mathcal{N}\big(\mathbf{z}_t; \sqrt{1 - \beta_t}\, \mathbf{z}_{t-1}, \, \beta_t \mathbf{I}\big),$$

with a fixed or learned variance schedule $\{\beta_t\}_{t=1}^{T}$. An important property of the forward noising process is that any marginal at step $t$ has a closed form (Ho et al., 2020): $q(\mathbf{z}_t \mid \mathbf{z}_0) = \mathcal{N}\big(\mathbf{z}_t; \sqrt{\bar{\alpha}_t}\, \mathbf{z}_0, \, (1 - \bar{\alpha}_t)\mathbf{I}\big)$, with $\alpha_t := 1 - \beta_t$ and $\bar{\alpha}_t := \prod_{s=1}^{t} \alpha_s$. Equivalently, any step $\mathbf{z}_t$ can be directly sampled from $\mathbf{z}_0$: $\mathbf{z}_t = \sqrt{\bar{\alpha}_t}\, \mathbf{z}_0 + \sqrt{1 - \bar{\alpha}_t}\, \boldsymbol{\epsilon}$, with $\boldsymbol{\epsilon} \sim \mathcal{N}(0, \mathbf{I})$. *The Reverse Process* is a Markov chain that iteratively denoises a sampled Gaussian noise to a clean data. Starting from $\mathbf{z}_T \sim \mathcal{N}(\mathbf{z}_T; 0, \mathbf{I})$, we learn a parameterized reverse process from latent $\mathbf{z}_T$ to clean data $\mathbf{z}_0$, as a chain of Gaussian transitions:

$$p_\phi(\mathbf{z}_{0:T}) = p(\mathbf{z}_T) \prod_{t=1}^{T} p_\phi(\mathbf{z}_{t-1} \mid \mathbf{z}_t), \qquad p_\phi(\mathbf{z}_{t-1} \mid \mathbf{z}_t) = \mathcal{N}\big(\mathbf{z}_{t-1}; \mu_\phi(\mathbf{z}_t, t), \Sigma_\phi \mathbf{I}\big).$$

The mean $\mu_\phi(\mathbf{z}_t, t)$ is primarily what we want to learn using a neural network (Ho et al., 2020). The variance $\Sigma_\phi$ can be either time-dependent constants or learnable (Nichol & Dhariwal, 2021). A function approximator $\boldsymbol{\epsilon}_\phi$ predicts the noise from $\mathbf{z}_t$ and sets: $\mu_\phi(\mathbf{z}_t, t) = \frac{1}{\sqrt{\alpha_t}}\big(\mathbf{z}_t - \frac{\beta_t}{\sqrt{1 - \bar{\alpha}_t}}\, \boldsymbol{\epsilon}_\phi(\mathbf{z}_t, t)\big)$. **Training** minimizes the $\ell_2$ loss between the true forward noise $\boldsymbol{\epsilon}$ and the predicted noise $\boldsymbol{\epsilon}_\phi(\mathbf{z}_t, t)$:

$$\mathcal{L}_{\text{diffusion}}(\boldsymbol{\phi}) = \mathbb{E}\big[\, \|\boldsymbol{\epsilon} - \boldsymbol{\epsilon}_\phi(\mathbf{z}_t, t)\|_2^2 \,\big]. \tag{2}$$

**Sampling/Guided Sampling.** During inference, sampling is performed by running the learned reverse process starting from Gaussian noise. Guided sampling augments this process with external signals (e.g., labels or features) to steer generation toward desired conditions (Dhariwal & Nichol, 2021).

**Inverse Diffusion Problem.** Given a degraded measurement $\mathbf{y} = \mathcal{D}(\mathbf{z}_0)$, $\mathcal{D}$ defines the degradation of the clean signal $\mathbf{z}_0$, the objective is to recover $\mathbf{z}_0$ by sampling from the posterior $p(\mathbf{z}_0 \mid \mathbf{y}) \propto p(\mathbf{y} \mid \mathbf{z}_0)\, p_\phi(\mathbf{z}_0)$, where $p_\phi(\mathbf{z}_0)$ is a diffusion prior learned from data (Kawar et al., 2022).

## 3 METHOD

### 3.1 PROBLEM FORMULATION

Let $\mathcal{D} = \{(\mathbf{x}_i, \mathbf{y}_i) \mid i = 1, \ldots, N\}$ denote a time series dataset with $N$ samples, where $\mathbf{x}_i \in \mathbb{R}^{M \times L}$ is a time series with $M$ channels and $L$ time steps, and $\mathbf{y}_i \in \mathbb{R}^C$ is the one-hot encoded label corresponding to the ground-truth class $c \in \{1, \ldots, C\}$. We henceforth write a generic sample as $\mathbf{x}$. We denote by $\mathbf{x}_e \in \mathbb{R}^{M \times e}$ a partially observed time series containing only the first $e < L$ time steps. While our main focus is on *timestep-wise partialness*, i.e., observing only a prefix of the sequence, we also evaluate the case where *channel-wise partialness* is present, with only a subset of channels $m < M$ observed, with $\mathbf{x}_{e,m} \in \mathbb{R}^{m \times e}$. The goal of this work is to learn student classifier $\mathcal{S}_{\boldsymbol{\theta}}$ that can effectively map early, partially observed inputs $\mathbf{x}_e$ to their corresponding labels $\mathbf{y}$, while leveraging the knowledge of a teacher $\mathcal{T}$ trained on the full-length sequences $\mathbf{x}$. We seek optimal student parameters $\boldsymbol{\theta}_*^{(e)}$ by minimizing

$$\mathcal{L}_{\text{Task}}(\boldsymbol{\theta}) = \mathbb{E}_{(\mathbf{x},\mathbf{y}) \sim \mathcal{D}}\big[\ell_{\text{CE}}(\mathcal{S}_{\boldsymbol{\theta}}(\mathbf{x}_e), \mathbf{y})\big], \quad \text{and} \quad \mathcal{L}_{\text{KD}}(\boldsymbol{\theta}) = \mathbb{E}_{\mathbf{x} \sim \mathcal{D}}\big[\ell(\phi_t(\mathbf{x}), \phi_s(\mathbf{x}_e; \boldsymbol{\theta}))\big].$$

Here, $\ell_{\text{CE}}$ is the cross-entropy loss, and $\phi_t, \phi_s$ are functions to be determined that capture teacher and student behavior on full and partial inputs, respectively. By minimizing the discrepancy measure $\ell(\cdot, \cdot)$, we encourage the student to behave on partially observed inputs as the teacher would on the full-length sequences.

### 3.2 GENERATIVE DIFFUSION PRIOR DISTILLATION

We write the student model as $\mathcal{S}_{\boldsymbol{\theta}} = \mathcal{S}_{\boldsymbol{\theta}}^{\text{head}} \circ \mathcal{S}_{\boldsymbol{\theta}}^{\text{feat}}$, where $\mathcal{S}_{\boldsymbol{\theta}}^{\text{feat}}$ denotes the mapping up to the feature extraction layer $k$, and $\mathcal{S}_{\boldsymbol{\theta}}^{\text{head}}$ denotes the subsequent mapping from these features to the final prediction. Training the student on the partial sequences $\mathcal{D}_e = \{(\mathbf{x}_e, \mathbf{y})\}^N$, we define $\mathcal{S}_{\boldsymbol{\theta}}^{\text{feat}}(\mathbf{x}_e) = \mathbf{z}_{\text{short}}$ as the intermediate feature of the partially observed input $\mathbf{x}_e$, referred to as the *short-context feature*. Let us assume there exists a feature $\mathbf{z}_{\text{long-ideal}}^*$, which encodes the optimal long-range information required for making accurate predictions from $\mathbf{x}_e$, as would be obtained if the model had access to and were ideally trained on full-length sequences. During training, our goal is to guide the student model such that it produces features $\mathbf{z}_{\text{short}}$ that resemble $\mathbf{z}_{\text{long-ideal}}^*$ as closely as possible, as if its predictions were informed by the full-length sequences, with the aim that this behavior generalizes to partial sequences at inference time.

**Teacher Knowledge as a Generative Prior.** We denote by $\mathbf{z}_{\text{long}}$ the features of a teacher model trained on the full-length sequences $\mathcal{D}$, referred to as the *long-context features*. To capture the teacher's knowledge, we train a diffusion model on $\mathbf{z}_{\text{long}}$ to approximate the distribution of long-context features, $p(\mathbf{z}_{\text{long}})$. We assume there exist features (possibly multiple) within the teacher's feature manifold that can provide useful hints of $\mathbf{z}_{\text{long-ideal}}^*$. We call these *hint features* and denote them by $\mathbf{z}_{\text{long-hint}} \sim p(\mathbf{z}_{\text{long}})$. We start by viewing $\mathbf{z}_{\text{short}}$ as a degraded or partial measurement of the underlying clean feature $\mathbf{z}_{\text{long-ideal}}^*$, which contains the optimal long-context knowledge and each $\mathbf{z}_{\text{long-hint}}$ as a valid approximation (or completion) of $\mathbf{z}_{\text{long-ideal}}^*$. The diffusion model trained on the teacher feature space serves as an effective prior $p_{\boldsymbol{\phi}}(\mathbf{z}_{\text{long}})$, capturing the statistics of plausible long-context features. Using this prior knowledge, our goal is to guide the student model to produce $\mathbf{z}_{\text{short}}$ that retains as much information as possible about its underlying clean feature $\mathbf{z}_{\text{long-ideal}}^*$. Accordingly, we aim for the student features to be minimally degraded relative to the hint features $\mathbf{z}_{\text{long-hint}}$, and thereby closer to $\mathbf{z}_{\text{long-ideal}}^*$.

To define the relationship between student features and hint features, we model the diffusion posterior sampler $\tilde{p}_{\text{diff}}(\mathbf{z}_{\text{long}} \mid \mathbf{z}_{\text{short}})$, where we utilize the pre-trained generative diffusion prior $p_{\boldsymbol{\phi}}(\mathbf{z}_{\text{long}})$ to search in the space of $\mathbf{z}_{\text{long}}$, for an optimal $\mathbf{z}_{\text{long}}$ that best matches $\mathbf{z}_{\text{short}}$, regarding $\mathbf{z}_{\text{short}}$ as degraded observation of $\mathbf{z}_{\text{long}}$. A posterior reconstruction sample $\hat{\mathbf{z}}_{\text{long}} \sim \tilde{p}_{\text{diff}}(\mathbf{z}_{\text{long}} \mid \mathbf{z}_{\text{short}})$ represents a plausible completion (clean signal) consistent with the partial information present in $\mathbf{z}_{\text{short}}$. We argue that, if the student produce features that preserve sufficient information of hint features, that is when $\mathbf{z}_{\text{short}}$ is minimally degraded to $\mathbf{z}_{\text{long-hint}}$, then student features should recover hint features as their posterior reconstruction samples. In other words, if $\mathbf{z}_{\text{short}}$ are sufficiently informative with valid long-context knowledge, then they should provide the right conditioning to recover one such representation, $\mathbf{z}_{\text{long-hint}}$, as their plausible completion:

$$\hat{\mathbf{z}}_{\text{long}}^* \sim \tilde{p}_{\text{diff}}(\mathbf{z}_{\text{long}} \mid \mathbf{z}_{\text{short}}; \boldsymbol{\theta}_*) \implies \hat{\mathbf{z}}_{\text{long}}^* \approx \mathbf{z}_{\text{long-hint}}, \tag{3}$$

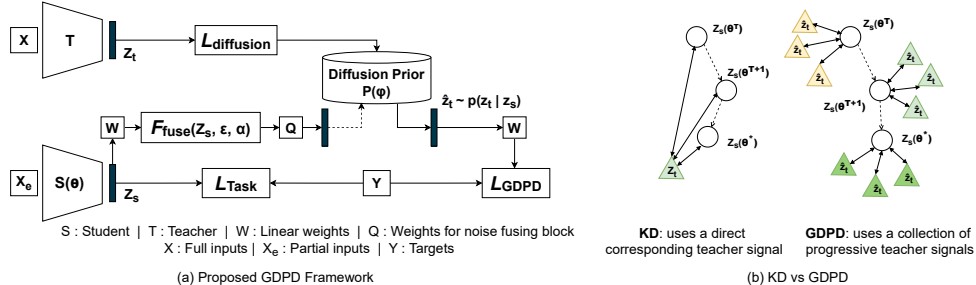

Figure 1: Depiction of (a) the proposed GDPD framework, and (b) a comparison of KD vs. GDPD. KD provides a single static teacher signal, whereas GDPD provides each student feature with a collection of diverse and progressive teacher signals over the course of training.

where $\boldsymbol{\theta}_*$ is the optimal student parameters, what we are after. Recall that hint features are functionally defined as teacher features that contain relevant long-context information necessary to assign the correct prediction to $\mathbf{x}_e$. We characterize hint features by this predictive property, which posterior reconstructions are trained to emulate under optimal student parameters. Therefore, during training we optimize the student features by constraining their posterior reconstructions to output the correct label:

$$\mathcal{L}_{\text{GDPD}}(\boldsymbol{\theta}) = \mathbb{E}_{(\mathbf{x},\mathbf{y})\sim\mathcal{D}}\Big[\ell_{\text{CE}}\Big(\mathcal{S}_{\boldsymbol{\theta}}^{\text{head}}(\hat{\mathbf{z}}_{\text{long}}^{(j)}),\mathbf{y}\Big)\Big], \quad \hat{\mathbf{z}}_{\text{long}} \sim \tilde{p}_{\text{diff}}\big(\mathbf{z}_{\text{long}} \mid \mathbf{z}_{\text{short}} = \mathcal{S}_{\boldsymbol{\theta}}^{\text{feat}}(\mathbf{x}_e)\big) \quad (4)$$

**Conditional Generation with Unconditional Prior.** To enable sampling from $p(\mathbf{z}_{\text{long}} \mid \mathbf{z}_{\text{short}})$ using an unconditional diffusion model, we adapt the reverse process to condition on $\mathbf{z}_{\text{short}}$ during inference, in line with guided sampling. Typically, guided sampling in inverse diffusion modifies the score function at each reverse step using an approximation of the likelihood gradients (Chung et al., 2023). In contrast to those settings, where the degraded measurements are fixed, our conditional signals are subject to optimization, which prompts us to require a simple and direct form of guidance from them (see Section A.4.1 for a detailed discussion). Therefore, we adopt a straightforward conditioning strategy by initializing the reverse diffusion process directly based on $\mathbf{z}_{\text{short}}$. Specifically we match each student feature $\mathbf{z}_{\text{short}}$ to the initial noisy step $T$ by fusing Gaussian noise, and use this initialization to start the reverse process:

$$\mathbf{z}_{\text{long},T} = \alpha\,\mathbf{z}_{\text{short}} + (1-\alpha)\,\boldsymbol{\epsilon}, \quad \boldsymbol{\epsilon} \sim \mathcal{N}(\mathbf{0},\mathbf{I}) \tag{5}$$

where $\alpha$ is the fusion weight between the two terms, which can be treated as a fixed hyperparameter or learned feature-wise during the distillation process. We choose to learn $\alpha$ during distillation, as this allows different features to be fused with different noise levels so that each is mapped appropriately to the initial noise step. With this initialization, reverse process is conditioned on $\mathbf{z}_{\text{short}}$, allowing the algorithm to explore the feature manifold, ideally staying close to the starting point $\mathbf{z}_{\text{short}}$, so that it converges to a plausible clean feature $\hat{\mathbf{z}}_{\text{long}}$ consistent to initialization. Over the course of optimization, posterior sampling connects different long-context features to a student feature (Equation (4)). This enables the student feature to evolve toward optimality by leveraging their collective knowledge, which can better approximates the knowledge of $\mathbf{z}_{\text{long-ideal}}^*$. An overview of this proposed method is illustrated in Figure 1 (a).

**Training.** Our student training proceeds in two phases, separated by a warm-up epoch $E_{\text{warm}}$. For $\text{ep} < E_{\text{warm}}$, we train only the diffusion prior on teacher features with the student initialized on the partial classification task. For $\text{ep} \geq E_{\text{warm}}$, the student is optimized to extract long-context knowledge using the learned diffusion prior.

$$\mathcal{L}_{\text{train}} = \begin{cases} \mathcal{L}_{\text{Task}}(\boldsymbol{\theta}) + \mathcal{L}_{\text{diffusion}}(\boldsymbol{\phi}), & \text{ep} < E_{\text{warm}}, \\ \lambda_{\text{Task}}\,\mathcal{L}_{\text{Task}}(\boldsymbol{\theta}) + \lambda_{\text{KD}}\,\mathcal{L}_{\text{GDPD}}(\boldsymbol{\theta}), & \text{ep} \geq E_{\text{warm}}. \end{cases} \tag{6}$$

### 3.3 "KNOWLEDGE" AS A DISTRIBUTION

**Conventional KD (deterministic / point knowledge).** KD treats the teacher signal that each student state (feature, soft label, or relation) should match as a *point target*. For a student state $\mathbf{Z}_s = \mathbf{z}_s$, the "knowledge" is taken to be a *single* teacher state: $\mathbf{k}^\star = \mathbf{z}_t$, with $\mathbf{z}_t$ the corresponding observed teacher state (equivalently, $P_{\mathbf{K}|\mathbf{Z}_s=\mathbf{z}_s} = \delta_{\mathbf{k}^\star}$). Supervision then enforces alignment of the student state with this single target, with a discrepancy loss: $\ell(\mathbf{z}_s; \boldsymbol{\theta}, \mathbf{k}^\star)$.

Common instances include:

$$\text{(Feature KD) } \mathbf{z}_s = f_s(\mathbf{x}; \boldsymbol{\theta}), \quad \mathbf{k}^\star = f_t(\mathbf{x}), \qquad \ell = \|f_s(\mathbf{x}; \boldsymbol{\theta}) - f_t(\mathbf{x})\|^2,$$

$$\text{(Logit KD) } \mathbf{z}_s = p_s(\cdot \mid \mathbf{x}; \boldsymbol{\theta}), \quad \mathbf{k}^\star = p_t(\cdot \mid \mathbf{x}), \qquad \ell = \text{KL}\big(p_t(\cdot \mid \mathbf{x}) \,\|\, p_s(\cdot \mid \mathbf{x}; \boldsymbol{\theta})\big),$$

$$\text{(Relational KD) } \mathbf{z}_s = r(f_s(\mathbf{x}; \boldsymbol{\theta}), f_s(\mathbf{x}'; \boldsymbol{\theta})), \quad \mathbf{k}^\star = r(f_t(\mathbf{x}), f_t(\mathbf{x}')), \qquad \ell = (\mathbf{z}_s - \mathbf{k}^\star)^2.$$

**GDPD (generative / distributional knowledge).** In contrast, GDPD provides a *distribution* of plausible teacher signals for each student state, consistent with what the student currently knows. Rather than treating knowledge as a single target, GDPD models it as a *distribution* from which the student can learn to sample in order to acquire optimal and diverse task-relevant knowledge. Formally, for a student state $\mathbf{z}_s$, the "knowledge" is a *distribution* over teacher states: $\mathbf{k} \sim p(\mathbf{K} \mid \mathbf{Z}_s = \mathbf{z}_s)$. More robust supervision can be defined as the *expected loss* (approximated by a Monte Carlo average over $J$ samples) under this distribution:

$$\mathbb{E}_{\mathbf{k} \sim p(\cdot|\mathbf{z}_s)}\big[\ell(\mathbf{z}_s; \boldsymbol{\theta}, \mathbf{k})\big] \quad \approx \quad \frac{1}{J}\sum_{j=1}^{J} \ell\big(\mathbf{z}_s; \boldsymbol{\theta}, \mathbf{k}^{(j)}\big), \quad \mathbf{k}^{(j)} \sim p(\cdot \mid \mathbf{z}_s). \tag{7}$$

Since each forward pass in GDPD explores a different noise trajectory, the stochasticity across training naturally covers multiple samples over time. Therefore, it is sufficient in practice to use $J = 1$. See ablation over $J$ in the Section 4.2.

### 3.4 HOW GDPD ADDRESSES FUNDAMENTAL KD CONCERNS, EXACERBATED IN FULL-TO-PARTIAL DISTILLATION

**How Does GDPD Transfer the Teacher Knowledge Effectively?** Since features derived from full and partial observations are not directly aligned in representation space, directly enforcing the partial-context student states $\mathbf{z}_s$ to match full-context teacher states $\mathbf{k}^* = \mathbf{z}_t$ as point targets can overwhelm the student. GDPD alleviates this gap by providing *dynamic and progressive teacher signals*. Each student state $\mathbf{z}_s$ samples its target teacher signal $\mathbf{k} \sim p(\mathbf{K} \mid \mathbf{Z}_s = \mathbf{z}_s; \boldsymbol{\theta}^t)$ from the teacher manifold, consistent with the student's current knowledge as reflected in $\mathbf{z}_s; \boldsymbol{\theta}^t$ during each forward pass. In this way, the teacher signal adapts over the course of training, allowing the student to progressively refine its ability to recover correct teacher features and ultimately absorb richer knowledge.

**How GDPD Provide Diverse Teacher Perspectives Using a Single Model?** When the student operates with degraded, incomplete features compared to the teacher, overfitting to a single teacher perspective can be risky, i.e., a single set of features (or logits) binds only one possible interpretation of the missing or noisy information. One possible way to create diverse perspectives from a single teacher is through stochastic diversity, i.e., generating multiple noise-perturbed variants of the same feature (Hossain et al., 2025). However, hand-designed perturbations may produce distant, irrelevant teacher signals, outside the meaningful teacher manifold. GDPD use diffusion models which naturally works by generating samples in the close vicinity of the target distribution (Chen et al., 2024). Unlike randomly perturbed feature variants, the diversity of diffusion-generated teacher signals in GDPD is controlled: they are not arbitrary but sampled as plausible completions of the student features, $\mathbf{k} \sim p(\mathbf{K} \mid \mathbf{Z}_s = \mathbf{z}_{\text{short}})$.

**How Does GDPD Transfer Faithful Knowledge?** When the teacher's training data and the distillation set differ, the knowledge from direct corresponding teacher signals $\mathbf{z}_t$ alone becomes very limited, making it difficult to faithfully replicate (predictive distributions of) the teacher (Stanton et al., 2021; Parchami-Araghi et al., 2024). In GDPD, as the student is trained toward optimal states, each student state interacts with a collection of teacher signals $\{\mathbf{k}^{(j)}\}; \mathbf{k}^{(j)} \sim p(\mathbf{K} \mid \mathbf{Z}_s = \mathbf{z}_s)$, which collectively construct comprehensive long-range knowledge for $\mathbf{z}_s$. See Figure 1 (b). Unlike

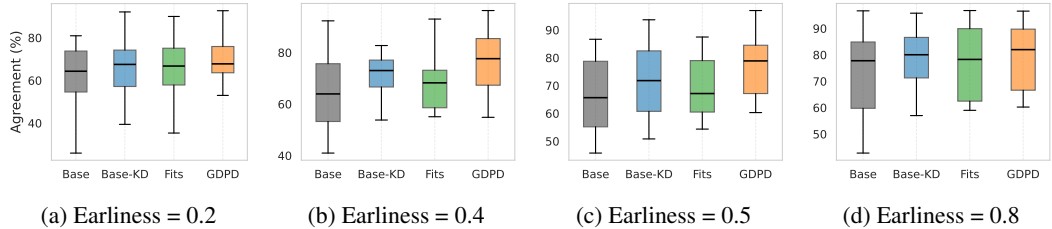

(a) Earliness = 0.2    (b) Earliness = 0.4    (c) Earliness = 0.5    (d) Earliness = 0.8

Figure 2: Fidelity comparisons across methods and earliness levels, with each boxplot summarizing 12 UCR datasets. Fidelity is measured as teacher–student top-1 agreement on the test set.

Table 1: Summary of performance across different earliness levels on 12 UCR datasets. Best values are in **bold**. Rows marked with ↓ indicate lower-is-better.

| | Earliness=0.2L | | | | Earliness=0.4L | | | | Earliness=0.6L | | | | Earliness=0.8L | | | |
|---|---|---|---|---|---|---|---|---|---|---|---|---|---|---|---|---|
| | Base | BaseKD | Fits | GDPD | Base | BaseKD | Fits | GDPD | Base | BaseKD | Fits | GDPD | Base | BaseKD | Fits | GDPD |
| Avg.AUC-PRC | 63.64 | 69.23 | 67.47 | **73.83** | 70.44 | 78.03 | 75.36 | **81.70** | 76.79 | 83.70 | 81.15 | **86.00** | 77.79 | 84.78 | 82.74 | **89.02** |
| Avg.Rank ↓ | 3.50 | 2.42 | 2.92 | **1.17** | 3.58 | 2.50 | 2.67 | **1.25** | 3.58 | 2.58 | 2.42 | **1.33** | 3.67 | 2.42 | 2.83 | **1.08** |
| Num.Top-1 | 0 | 2 | 0 | **10** | 0 | 1 | 1 | **10** | 0 | 3 | 2 | **8** | 0 | 0 | 1 | **11** |
| Wins/Draws | 12 | 10 | 12 | – | 12 | 11 | 10 | – | 12 | 10 | 10 | – | 12 | 12 | 11 | – |
| Losses ↓ | 0 | 2 | 0 | – | 0 | 1 | 2 | – | 0 | 2 | 2 | – | 0 | 0 | 1 | – |

the supervision from a single corresponding teacher signal $\mathbf{k}^* = \mathbf{z}_t$, this collection better reveals the statistical structure of teacher features (class separability, geometric relationships between features). We validate faithful knowledge empirically in Section 4.1.

## 4 EXPERIMENTS

To reach generalizable conclusions across partialness levels, architectures, datasets, and runs, we evaluate multiple settings. Experiments are conducted on the UCR univariate (Dau et al., 2019), UEA multivariate (Bagnall et al., 2018), and a real-world PhysioNet mortality dataset (Silva et al., 2012). Notation $Net1 \rightarrow Net2$ denotes teacher–student distillation. All students use the same training protocol, and results are averaged over five runs. Full experimental setup details are in Section A.2.

### 4.1 MAIN RESULTS

**GDPD is Effective Across Varying Degrees of Partialness.** We generate time series with varying earliness by truncating at $e \in \{0.2L, 0.4L, 0.5L, 0.6L, 0.8L, L\}$, where $L$ is the full length. The teacher is trained on full-length series, while students are trained on truncated series using KD from logits (Base-KD) (Hinton et al., 2015), features (Fits) (Romero et al., 2015), and GDPD, along with a baseline student (Base) trained without distillation. Results for LSTM3-100 $\rightarrow$ LSTM3-100 on 12 UCR datasets are shown in Table 1 (and Table 19). At each earliness level, distilled students achieve higher AUC-PRC and lower rank than the Base student, showing that full-context teacher knowledge improves partial classification. GDPD attains the best AUC-PRC and rank, winning on over 80% of datasets, demonstrating its effectiveness over direct feature- and logit-KD across partialness levels.

**GDPD Outperforms Existing KD Variants.** To further compare GDPD with different KD variants, we construct students using RKD (Park et al., 2019), Attention (Zagoruyko & Komodakis, 2016), DKD (Zhao et al., 2022), DT2W (Qiao et al., 2023), VID (Ahn et al., 2019), PKT (Passalis et al., 2020), TeKAP (Hossain et al., 2025), TTM (Zheng & Yang, 2024), Base-KD, Fits, and Base. Performance is evaluated at partialness $e = 0.5L$ on 12 UCR datasets for LSTM3-100 $\rightarrow$ LSTM3-100 (Table 2). All students improve over the Base, showing that they benefit from long-context knowledge. GDPD outperforms all methods, achieving top-3 performance on 80% datasets with an average rank of 2.25, while no other method achieves a rank close to 4.

**GDPD Improves Student Fidelity** Beyond generalization across earliness levels, we also assess whether GDPD yields students with *higher fidelity*. Figure 2 summarizes student fidelity, measured as *average top-1 agreement* over 12 UCR datasets for LSTM3-100 $\rightarrow$ LSTM3-100 at different earliness levels. This metric is the percentage of instances where the student's top-1 prediction (from

Table 2: Comparison of distillation methods at earliness $e = 0.5L$ on 12 UCR datasets. Test-fidelity is reported as average top-1 agreement. Rows marked with ↓ indicate lower-is-better.

| | Base | Base-KD | Fits | VID | DKD | Attention | DT2W | RKD | RKD-Angle | PKT | TeKAP | TTM | GDPD |
|---|---|---|---|---|---|---|---|---|---|---|---|---|---|
| Avg. AUC-PRC | 71.6 | 78.14 | 77.28 | 77.19 | 74.29 | 75.47 | 55.32 | 79.89 | 79.26 | 76.86 | 76.54 | 78.05 | **84.64** |
| Test-Fidelity | 66.75 | 72.13 | 69.39 | 71.32 | 69.36 | 69.20 | 52.59 | 74.34 | 71.22 | 71.05 | 69.53 | 70.24 | **77.58** |
| Avg. Rank ↓ | 9.67 | 5.83 | 7.58 | 6.33 | 8.25 | 7.25 | 11.42 | 6.33 | 6.42 | 6.67 | 6.75 | 6.17 | **2.25** |
| Num. Top-1 | 0 | 2 | 0 | 0 | 0 | 0 | 0 | 2 | 2 | 0 | 1 | 0 | **5** |
| Num. Top-3 | 0 | 5 | 2 | 1 | 1 | 2 | 0 | 4 | 4 | 3 | 2 | 2 | **10** |
| Wins/Draws | 12 | 10 | 12 | 11 | 11 | 11 | 12 | 9 | 9 | 10 | 11 | 11 | – |
| Losses | 0 | 1 | 0 | 1 | 1 | 1 | 0 | 3 | 3 | 2 | 1 | 1 | – |

Table 3: Results on 12 multivariate datasets under: 1) time-wise ($e = 0.5L$), and 2) time+channel ($e = 0.5L, m = 0.5M$) partialness. Each cell reports Avg. AUC-PRC / Avg. Rank / #Top-1 wins.

| | Inception55-32 → Inception55-32 | | LSTM3-100 → LSTM3-100 | |
|---|---|---|---|---|
| | Time-wise | Time+Channel | Time-wise | Time+Channel |
| Base | 86.22 / 3.25 / 0 | 76.58 / 2.92 / 0 | 79.41 / 3.50 / 0 | 68.75 / 3.42 / 0 |
| Base-KD | 88.77 / 2.00 / 3 | 77.92 / 2.08 / 3 | 81.96 / 2.50 / 3 | 71.11 / 2.75 / 1 |
| Fits | 85.40 / 3.17 / 2 | 75.34 / 3.42 / 1 | 81.95 / 2.42 / 2 | 71.69 / 2.33 / 2 |
| GDPD | **90.82 / 1.33 / 9** | **80.24 / 1.33 / 10** | **83.87 / 1.58 / 7** | **73.60 / 1.42 / 9** |

Table 4: Model compression results under two earliness levels ($e = 0.5L, L$) and two compression targets. Each cell reports Avg. AUC-PRC / Avg. Rank / Num. Top-1 wins across 12 UCR datasets.

| | LSTM3-100 → LSTM1-8 | | LSTM3-100 → LSTM2-32 | |
|---|---|---|---|---|
| | $e = 0.5L$ | $e = L$ | $e = 0.5L$ | $e = L$ |
| Base | 58.58 / 3.42 / 0 | 74.92 / 3.33 / 0 | 72.98 / 3.33 / 0 | 84.34 / 3.67 / 0 |
| Base-KD | 61.97 / 2.33 / 1 | 77.62 / 2.33 / 3 | 74.92 / 2.58 / 2 | 88.11 / 2.17 / 4 |
| Fits | 59.84 / 2.67 / 0 | 76.49 / 2.83 / 2 | 78.26 / 2.75 / 1 | 85.93 / 2.75 / 1 |
| GDPD | **72.76 / 1.17 / 11** | **78.83 / 1.42 / 7** | **83.67 / 1.33 / 9** | **89.84 / 1.25 / 9** |

partial data) matches the teacher's top-1 prediction (from the full sequence), directly quantifying how well the student replicates the teacher's behavior (Stanton et al., 2021). In all settings, GDPD students achieve higher fidelity than baseline KD, demonstrating its effectiveness in faithfully transferring desirable teacher behavior under different degrees of partialness.

**GDPD is Robust with Channel-wise Partialness.** We evaluate GDPD's robustness under channel-wise partialness using 12 UEA multivariate datasets with two settings: 1) time-wise partialness only ($e = 0.5L$), and 2) combined time- and channel-wise partialness ($e = 0.5L, m = 0.5M$), where half the channels are removed (Table 3). GDPD consistently achieves the highest AUC-PRC, lowest average ranks, and most top-1 wins, confirming its robustness to channel-wise partialness.

**GDPD is Effective in Model Compression.** We evaluate GDPD's effectiveness for model compression under two scenarios: 1) compression with partialness ($e = 0.5L$) and 2) compression only ($e = L$). Results for two compression targets in Table 4 show GDPD consistently achieves the highest AUC-PRC, lowest rank, and most wins, confirming its effectiveness for compressed students.

**GDPD Provides Effective Self-Distillation.** When teacher and student architectures are identical and no partialness is involved ($e = L$), our experiments correspond to self-distillation (Pham et al., 2022). We evaluate GDPD under this setting in Table 5. GDPD achieves higher AUC-PRC and lower rank than the teacher, surpassing them on most datasets. GDPD proves more effective for self-distillation than vanilla KD. **Justification for improvements:** Even without a teacher–student representational gap, GDPD exposes each student feature to multiple vicinal features (analogous to feature augmentation), which helps the model generalize better while avoiding overconfidence.

**GDPD Across Different Network Architectures.** We also evaluate GDPD under 1) similar teacher–student and 2) cross-architecture distillation. GDPD consistently achieves the highest performance gains, demonstrating effectiveness in both settings (see Table 18).

**GDPD Transfers Long-context Knowledge.** To assess whether GDPD students learn representations that generalize from full-context teacher knowledge, we evaluate representation transferability to the *suffix* (last $0.5L$) of each time series. We use two protocols: 1) **Linear probe on frozen backbone:** Train the student on *prefix* data (first $0.5L$), freeze its backbone, then train a *linear* clas-

Table 5: Self-distillation results under three network architectures ($e = L$). Each cell reports Avg. AUC-PRC / Avg. Rank across 12 UCR datasets.

| | LSTM3-100 $\rightarrow$ LSTM3-100 | Inception55-32 $\rightarrow$ Inception55-32 | ResNet32-64 $\rightarrow$ ResNet32-64 |
|---|---|---|---|
| Teacher | 87.32 / 3.33 | 96.96 / 2.83 | 98.43 / 2.33 |
| Base-KD | 89.29 / 2.08 | 97.80 / 1.50 | 98.52 / 2.25 |
| Fits | 88.73 / 2.75 | 93.90 / 3.92 | 98.55 / 2.33 |
| GDPD | **91.21 / 1.58** | **97.97 / 1.25** | **98.58 / 1.25** |

Table 6: Transferability from prefix-trained students to suffix inputs on the StarLightCurves dataset. We report AUC-PRC; best values per row are in **bold**.

| | Base | Base-KD | Fits | GDPD |
|---|---|---|---|---|
| Linear-probe | 65.57 | 74.86 | 65.74 | **76.88** |
| Zero-shot | 35.21 | 66.21 | 48.05 | **66.70** |

Table 7: Results on the PhysioNet case study in terms of AUC-ROC, AUC-PRC, and Accuracy. Four settings under $e = 0.5L$ are considered: (1) main-task distillation, (2) downstream task (survival $\geq 100$ prediction), (3) channel-wise partialness, and (4) balanced-teacher to imbalanced-student.

| | In-hospital Mortality ($e = 0.5L$) | | | Survival $\geq 100$ ($e = 0.5L$) | | | Channel Partialness ($e = 0.5L, m = 0.5M$) | | | Balanced $\rightarrow$ Imbalanced ($e = 0.5L$) | | |
|---|---|---|---|---|---|---|---|---|---|---|---|---|
| | ROC | PRC | Acc. | ROC | PRC | Acc. | ROC | PRC | Acc. | ROC | PRC | Acc. |
| Teacher | 76.04 | 76.12 | 70.72 | 76.04 | 76.12 | 70.72 | 76.04 | 76.12 | 70.72 | 76.04 | 76.12 | 70.72 |
| Base | 70.21 | 68.70 | 65.41 | 67.80 | 65.58 | 62.78 | 60.38 | 59.52 | 55.50 | 75.96 | 65.44 | 72.18 |
| Fits | 73.54 | 72.50 | 67.03 | 68.61 | 66.61 | 63.48 | 59.84 | 59.79 | 56.04 | 76.84 | 65.66 | **86.18** |
| GDPD | **74.45** | **73.74** | **68.56** | **70.52** | **69.74** | **65.13** | **61.31** | **62.18** | **57.40** | **76.88** | **65.84** | 86.10 |

sifier on features extracted from *suffix* inputs. This tests whether prefix-learned representations are linearly useful for classifying suffix inputs. 2) **Zero-shot suffix evaluation:** Evaluate the frozen prefix-trained student (backbone + original head) directly on *suffix* inputs without any additional training. Results on the StarLightCurves dataset for LSTM3-100 $\rightarrow$ LSTM3-100 are presented in Table 6. These results indicate that representations learned with GDPD exhibit strong transferability: GDPD achieves the best linear-probe and zero-shot performance on suffix inputs, evidencing that it effectively acquires and transfers long-context temporal knowledge.

**GDPD Degrades Gracefully under Weak Teacher Supervision.** Under progressively degraded teacher supervision, GDPD consistently achieves the highest performance and the slowest decline (Section A.3.5).

**Real-World Case Study: Predicting In-Hospital Mortality.** The PhysioNet Silva et al. (2012) dataset contains electronic health records from ICU patients. The main task is to predict in-hospital mortality using first 48 hours recordings after admission. We also derive an auxiliary downstream task, *survival $\geq 100$ prediction*, to evaluate cross-task distillation (see Section A.2.1). We train the teacher on the main task using a balanced set of full-length data. In the first scenario, students are trained on the same task with partialness ($e = 0.5L$). We then consider three additional scenarios introducing further teacher–student heterogeneity: 1) training students on a downstream task (survival $\geq 100$ prediction) with $e = 0.5L$; 2) training under channel-wise partialness ($e = 0.5L, m = 0.5M$); and 3) training on an imbalanced dataset with $e = 0.5L$. Table 7 indicates GDPD consistently outperforms Base and feature-KD across time- and channel-wise partialness, cross-task distillation, and when the class distribution of the distillation set differs from the teacher's training data, highlighting its robustness under heterogeneous real-world conditions. Cross-task gains show GDPD extracts task-relevant knowledge more effectively than direct feature matching.

## 4.2 ABLATIONS AND HYPERPARAMETER STUDY

To gain insights into the role of each component in Equation (6), we conduct an ablation study on the *StarLightCurves* dataset for LSTM3-100 $\rightarrow$ LSTM3-100 with earliness set to $e = 0.5L$.

**Ablation on Phase Scheduling.** Under disabled warm-up and diffusion training ($E_{\text{warm}} = 0$, $\mathcal{L}_{\text{diffusion}}(\phi) = 0$), freezing $\phi$ yields 83.12 AUC-PRC and training without freezing yields 85.56, both near the Base (83.90), indicating GDPD provides no benefit without proper diffusion prior training. With warm-up disabled, jointly training the diffusion prior ($\mathcal{L}_{\text{diffusion}} + \mathcal{L}_{\text{GDPD}}$)

Table 8: Effect of warm-up epochs ($E_{\mathrm{warm}}$) on GDPD performance (AUC-PRC).

| $E_{\mathrm{warm}}$ | 0 (frozen $\phi$) | 0 (unfrozen $\phi$) | 0 (joint) | 10 | 50 | 100 | 200 | 300 | 400 | 600 |
|---|---|---|---|---|---|---|---|---|---|---|
| AUC-PRC | 83.12 | 85.56 | 89.72 | 87.4 | 97.13 | 95.31 | 95.83 | **97.26** | 96.78 | 83.90 |

yields 89.72 AUC-PRC, suggesting coupling helps even without a warm-up. Next, we sweep $E_{\mathrm{warm}}$ in Table 8. At $E_{\mathrm{warm}} = 600 =$ total epochs, training reduces to the task loss only, collapsing to the Base (83.90). The best results emerge when the warm-up occupies roughly half of the training epochs (e.g., 97.26 at $E_{\mathrm{warm}} = 300$). These findings show that 1) the diffusion prior is essential, 2) phase-wise training improves GDPD, and 3) mid-range warm-up durations yield the highest gains.

**Ablation on Loss Terms.** Setting $\lambda_{\mathrm{KD}} = 0$ yields the Base (83.90). Using only the GDPD signal ($\lambda_{\mathrm{Task}} = 0$) gives limited improvement (85.72), while combination ($\lambda_{\mathrm{Task}} = 1$, $\lambda_{\mathrm{KD}} = 1$) achieves the best result (97.26). With $\lambda_{\mathrm{Task}} = 1$, we sweep $\lambda_{\mathrm{KD}}$ in Table 13, where any non-zero GDPD contribution improves over $\lambda_{\mathrm{KD}} = 0$. To verify that GDPD drives this gain, we replace $\mathcal{L}_{\mathrm{GDPD}}$ (Equation (6)) with logit-KD, which gives 94.51, close to Base-KD (95.20) but inferior to GDPD. Substituting feature-KD yields 83.81, similar to Fits (81.87). This suggests that, instead of matching a single static teacher signal, GDPD's diverse and progressive signals drives performance gains. We further validate this by increasing diversity, estimating $\mathcal{L}_{\mathrm{GDPD}}$ with multiple posterior samples $J = 1, 2, 3, 4, 5$ (Equation (7)), which yield 97.26, 97.34, 97.13, 97.87, and 97.78, respectively. The gain for $J > 1$ can be attributed to the additional diversity introduced within each mini-batch. Since $J = 1$ already achieves strong performance, we adopt it for a better efficiency–accuracy trade-off. We also ablate alternative GDPD implementations that reduce diversity in Section A.3.1.
**Ablation on Diffusion Controls** are provided in Section A.3.1.

**Computational Cost.** GDPD incurs only modest training-time overhead (0.24 s/epoch over Base-KD) and does not affect inference. Its computational cost is comparable to established feature KD (RKD, VID), and memory footprint is far below memory-intensive methods (RKD). Detailed wall-clock and step-level analyses are provided in Section A.3.2.

## 5 CONCLUSION

This paper proposes a novel KD framework for efficient knowledge transfer to bridge the generalization gap from partial data. In conventional KD, directly matching a single set of teacher features can result in incomprehensible knowledge due to data gaps, limited and brittle knowledge tied to one perspective, and unfaithful knowledge from training–distillation set differences. To address this, we propose capturing teacher knowledge as a generative diffusion prior that serves as a reservoir from which the student can progressively sample diverse and faithful knowledge. We conduct extensive evaluations of GDPD across partialness levels and distillation settings, demonstrating consistent improvements over existing KD approaches on benchmark datasets. Additionally, we validate GDPD on a real-world dataset under challenging heterogeneous conditions. This paper opens a new research direction by proposing teacher knowledge as an effective form of generative prior.

## ACKNOWLEDGMENTS

This research was supported by The University of Melbourne's Research Computing Services and the Petascale Campus Initiative. We also gratefully acknowledge the authors and maintainers of the UCR, UEA, and PhysioNet time series repositories for making their datasets publicly available.

## REPRODUCIBILITY STATEMENT

The source code for all models, training scripts, and experiments is available at `https://github.com/hewadehigaha/GDPD_ICLR26`. Details of datasets, preprocessing steps, model configurations, and training procedures are provided in the main text (Section 4) and Appendix (Section A.2). Additional results are included in the supplementary materials to further support reproducibility.

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

# A APPENDIX

## A.1 RELATED WORK

**Knowledge Distillation.** KD, introduced by Buciluǎ et al. (2006); Hinton et al. (2015), demonstrates that smaller models can achieve comparable or superior performance through knowledge transfer from larger models. This process involves matching the teacher's softened logits with those of the student, adjusted by a temperature hyper-parameter to amplify the contribution of negative logits. Incorporating intermediate feature representations alongside final-layer logits has further improved performance, establishing state-of-the-art results (Romero et al., 2015; Zagoruyko & Komodakis, 2016; Ahn et al., 2019; Park et al., 2019).

However, recent works observe that KD often fails to meet its conventional promise due to several concerns: 1) Distillation may not transfer teacher knowledge effectively because of the capacity or architectural gap between teacher and student. To address this, recent studies have proposed intermediate "teacher–assistant" models (Mirzadeh et al., 2020; Son et al., 2021) and student-friendly teacher training (Park et al., 2021; Rao et al., 2023; Cho & Hariharan, 2019). 2) Knowledge from a single teacher is often not diverse enough, as it reflects only one perspective. To address this, existing works promote diversity through teacher ensembles (Allen-Zhu & Li, 2020; You et al., 2017) or mutual supervision among student ensembles (Zhang et al., 2018; Furlanello et al., 2018). More recently, Hossain et al. (2025) generate multiple augmented teacher views from a single model by perturbing features with random noise, thereby increasing knowledge diversity while avoiding the cost of retraining multiple models. 3) Knowledge is not always faithful: recent works observe that KD often transfers limited knowledge, leading students to learn predictive distributions very different from their teachers, which hinders their safe substitution (Stanton et al., 2021; Lamb et al., 2023). To mitigate this, recent studies have proposed transferring properties beyond direct logits or features, which has been shown to improve student fidelity (Parchami-Araghi et al., 2024; Lamb et al., 2023). When distilling knowledge from a teacher trained on full-length data to a student operating on partial data, all of these concerns are further exacerbated by the additional training–distillation data mismatch (Stanton et al., 2021).

**Partial Time-Series Classification** There is a related but distinct line of research called early time-series classification (eTSC) (Mori et al., 2017; Schäfer & Leser, 2020), which aims to predict as early as possible without observing the full sequence. The model processes a growing prefix and decides at each step whether to predict or wait, trading off earliness and accuracy. In contrast, classification with partial time series assumes only a fixed prefix is available by constraint (e.g., latency, cost, or sensor dropout). Unlike eTSC, there is no option to defer prediction, and the model must classify directly from incomplete and ambiguous data. This work investigates whether a model operating on partial time series can benefit from the generalization of a model trained on full-length sequences, which may have learned robust representations across time. While none of the existing works specifically address prefix-based partialness, a few approaches mitigate the generalization gap from channel-wise partialness by distilling multi-lead ECG classifiers to single-lead models (Sepahvand & Abdali-Mohammadi, 2022; Chauhan et al., 2022). However, these application-specific methods rely on direct feature- or logit-level KD and overlook the training–distillation data mismatch. This work address this problem in a broad and innovative manner by modeling teacher–student feature relations as degraded–to–clean counterparts and leveraging a generative prior to recover long-range temporal discriminative cues.

## A.2 EXPERIMENTAL SETUP

**Teacher and Student Models** In our experiments, we primarily use a Long Short-Term Memory (LSTM) network (Hochreiter, 1997) (built upon recurrent blocks), ResNet (Wang et al., 2017) (a network primarily composed of convolutional layers), and an InceptionTime network (Ismail Fawaz et al., 2020), which is among the current state-of-the-art for TSC. For experiments involving model compression, we construct smaller variants of LSTM under different compression levels by varying the number of layers and output dimensions. The total number of parameters, model sizes, and network configurations for all constructed models are summarized in Table 9.

Table 9: Configuration of networks used for student and teacher models. The output dimension indicates the hidden size for the LSTM and the output dimension of the first convolutional layer for InceptionTime (Ismail Fawaz et al., 2020) and ResNet (Wang et al., 2017).

| Network | Num. Layers | Output Dim. | Total Param. | Model Size (MB) |
|---|---|---|---|---|
| Inception55-32 | 55 | 32 | 978440 | 0.9361 |
| Resnet32-64 | 32 | 64 | 2016008 | 1.9315 |
| LSTM3-100 | 3 | 100 | 812008 | 0.7744 |
| LSTM2-32 | 2 | 32 | 51976 | 0.0496 |
| LSTM1-8 | 1 | 8 | 1480 | 0.0014 |

**Datasets.** We conducted our experiments using 12 univariate time-series datasets from the UCR-2015 archive (Dau et al., 2019) and 12 multivariate datasets from the UEA archive (Bagnall et al., 2018). Details of the selected datasets are provided in Table 10 and Table 11, respectively. All series were standardized to length 100 via linear interpolation, z-normalized, and evaluated with the original train/test split with 20% validation.

Table 10: Summary of univariate UCR benchmark datasets used in our experiments.

| Dataset | Type | Train | Test | Variables (M) | Length (L) | Categories (C) |
|---|---|---|---|---|---|---|
| CBF | Simulated | 30 | 900 | 1 | 128 | 3 |
| Coffee | Spectro | 28 | 28 | 1 | 286 | 2 |
| ECG200 | ECG | 100 | 100 | 1 | 96 | 2 |
| ECGFiveDays | ECG | 23 | 861 | 1 | 136 | 2 |
| GunPoint | Motion | 50 | 150 | 1 | 150 | 2 |
| FaceAll | Image | 560 | 1690 | 1 | 131 | 14 |
| ItalyPowerDemand | Sensor | 67 | 1029 | 1 | 24 | 2 |
| NonInvasiveFetalECGThorax1 | ECG | 1800 | 1965 | 1 | 750 | 42 |
| StarLightCurves | Sensor | 1000 | 8236 | 1 | 1024 | 3 |
| SyntheticControl | Simulated | 300 | 300 | 1 | 60 | 6 |
| Trace | Sensor | 100 | 100 | 1 | 275 | 4 |
| TwoLeadECG | ECG | 23 | 1139 | 1 | 82 | 2 |

Table 11: Summary of multivariate UEA benchmark datasets used in our experiments.

| Dataset | Train | Test | Variables (M) | Length (L) | Categories (C) |
|---|---|---|---|---|---|
| ArticularyWordRecognition | 275 | 300 | 9 | 144 | 25 |
| BasicMotions | 40 | 40 | 6 | 100 | 4 |
| Cricket | 108 | 72 | 6 | 1197 | 12 |
| ERing | 30 | 270 | 4 | 65 | 6 |
| JapaneseVowels | 270 | 370 | 12 | 29 | 9 |
| Libras | 180 | 180 | 2 | 45 | 15 |
| NATOPS | 180 | 180 | 24 | 51 | 6 |
| PenDigits | 7494 | 3498 | 2 | 8 | 10 |
| PEMS-SF | 267 | 173 | 963 | 144 | 7 |
| RacketSports | 151 | 152 | 6 | 30 | 4 |
| SelfRegulationSCP1 | 268 | 293 | 6 | 896 | 2 |
| UWaveGestureLibrary | 120 | 320 | 3 | 315 | 8 |

**Implementation Details.** We select the best teacher model from five random initializations based on the validation area under the precision-recall curve (AUC-PRC) (Wang et al., 2017). All students involving KD are trained using a combination of the task loss and the distillation loss:

$$\mathcal{L}_{\text{train}}(\boldsymbol{\theta}) = \lambda_{\text{Task}} \mathcal{L}_{\text{Task}}(\boldsymbol{\theta}) + \lambda_{\text{KD}} \mathcal{L}_{\text{KD}}(\boldsymbol{\theta}),$$

where $\lambda_{\text{Task}}$ and $\lambda_{\text{KD}}$ determine the contributions of the classification loss $\mathcal{L}_{\text{Task}}$ (cross-entropy) and the distillation loss $\mathcal{L}_{\text{KD}}$, respectively. For all experiments, $\lambda_{\text{Task}}$ is fixed at 1, while $\lambda_{\text{KD}}$ is optimized via grid search over $\{0.1, 1, 10\}$. Models are implemented in PyTorch (Paszke et al., 2019) and trained with the Adam optimizer using a batch size of 64 for a maximum of 600 epochs, with the best weights selected based on validation loss. For GDPD students, the warm-up epoch is set to $E_{warm} = 300, 350$, nearly half of the total epochs. A learning rate decay of 0.5 is applied at epochs 25, 30, and 35, with initial learning rates set to 0.01 for the LSTM3-100 and LSTM2-32 models, and 0.1 for the other models. All student results are reported as the average over five runs with different random initializations.

**Implementation of GDPD.** We adopt a lightweight DDIM implementation together with the noise fusing block proposed by Huang et al. (2023) for diffusion prior training, using a total of 1000 diffusion steps. All the hyperparameters used to implement GDPD are listed in Table 12.

Table 12: Default hyperparameters used for implementing GDPD.

| Parameter | Value |
|---|---|
| Diffusion steps ($T$) | 1000 |
| Number of NFEs (sampling steps) | 5 |
| Knowledge distillation weight ($\lambda_{\text{KD}}$) | Best among {0.1, 1, 10} |
| Task loss weight ($\lambda_{\text{Task}}$) | 1.0 |
| Number of posterior samples ($J$) | 1 |
| Total training epochs | 600 |
| Warm-up epochs ($E_{\text{warm}}$) | Best among {300, 350} |
| Batch size | 64 |
| Optimizer | Adam |

**Evaluation Metrics.** Model performance was primarily evaluated using area under the receiver operating characteristic curve (AUC-ROC), average AUC-PRC, and accuracy on the test set. We adopt a metric from Stanton et al. (2021) to measure model fidelity: the average agreement between the student's and teacher's top-1 predictions:

$$\text{Average Top-1 Agreement} = \frac{1}{N} \sum_{i=1}^{N} \mathbb{1}\left(y_{t,i} = y_{s,i}\right),$$

A win/draw/loss calculation was employed, where a model 'wins' on a dataset, if it achieves the highest AUC-PRC. We prioritized AUC-PRC over other metrics due to its robustness to class imbalance.

The reported metrics in Table 1 and Table 2 are:

- **Avg. AUC-PRC:** The average AUC-PRC across all datasets.
- **Avg. Test-Fidelity:** The average teacher-student agreement across all datasets.
- **Avg. Rank:** The average ranking of a method compared to all baselines (lower is better).
- **Num. Top-1:** The number of datasets where the method achieves the highest performance (AUC-PRC) among all baselines.
- **Num. Top-3:** The number of datasets where the method ranks within the top three in performance.
- **Wins/Draws:** The number of datasets where GDPD achieves equal or better performance compared to all baselines.
- **Losses:** The number of datasets where GDPD underperforms compared to baselines.

### A.2.1 PREDICTING IN-HOSPITAL MORTALITY ON PHISONET DATA

The PhysioNet Silva et al. (2012) dataset contains medical records collected during the first 48 hours after patients were admitted to an intensive care unit. A total of 37 variables were observed one or more times for each patient, along with labels indicating length of stay (days), survival (days), and in-hospital death. Omitting categorical variables, we use 11 time series variables:

- **DiasABP:** Invasive diastolic arterial blood pressure (mmHg)
- **FiO2:** Fractional inspired oxygen (0–1)
- **HR:** Heart rate (bpm)
- **MAP:** Invasive mean arterial blood pressure (mmHg)
- **NIMAP:** Non-invasive mean arterial blood pressure (mmHg)
- **SaO2:** Oxygen saturation in hemoglobin (%)
- **RespRate:** Respiration rate (bpm)
- **NISysABP:** Non-invasive systolic arterial blood pressure (mmHg)

- **SysABP:** Invasive systolic arterial blood pressure (mmHg)
- **Glucose:** Serum glucose (mg/dL)
- **Temp:** Temperature (°C)

Therefore each sample is a time series with $M = 11$ variables and $L = 48$ timesteps. We impute missing values using forward/backward filling, with train-set feature means for entirely missing channels.

**Downstream Task.** In addition to the in-hospital mortality classification task (label = in-hospital death), we define a downstream task from the multi-label annotations: survival $\geq 100$ days prediction (label = 1 if survival == -1 or survival $\geq 100$; else 0), to evaluate performance under cross-task distillation.

## A.3 ADDITIONAL RESULTS

### A.3.1 FURTHER ABLATION STUDIES

**Ablation on Loss Terms.** To further verify that the diversity of GDPD's teacher signals drives the gain, we modify $\mathcal{L}_{\mathrm{GDPD}}$ (Equation (4)) as $\mathbb{E}_{(\mathbf{x},\mathbf{y})\sim\mathcal{D}}\left[\left\|\hat{\mathbf{z}}_{\mathrm{long}}^{(1)} - \mathbf{z}_{\mathrm{long}}^*\right\|^2\right]$, where posterior reconstruction samples are constrained to match the corresponding direct teacher feature $\mathbf{z}_{\mathrm{long}}^*$ of the same training sample. This reduces knowledge diversity, forcing exact reconstruction, with a reduced result of 96.21 that confirms the limitation.

We present ablation for Distillation ratio ($\lambda_{\mathrm{KD}}$) in Table 13.

Table 13: Ablation of distillation ratio: GDPD performance measured in terms of AUC-PRC.

| Distillation ratio ($\lambda_{\mathrm{KD}}$) | 0 | 0.01 | 0.1 | 0.5 | 1 | 10 | 100 |
|---|---|---|---|---|---|---|---|
| AUC-PRC | 83.90 | 86.41 | 91.91 | 93.72 | **97.26** | 91.87 | 88.61 |

**Ablation on Diffusion Controls.** We ablate the number of forward diffusion steps $T$ in $\mathcal{L}_{\mathrm{diffusion}}(\phi)$ with $\{100, 500, 800, 1000\}$, obtaining 93.29, 95.33, 92.49, and 97.26, and set $T = 1000$ in our experiments. Following Huang et al. (2023), we use DDIM (Song et al., 2020a), which accelerates denoising compared to early diffusion models and allows sampling with far fewer score function evaluations (NFEs) $\ll T$. We ablate the NFEs in the $\mathcal{L}_{\mathrm{GDPD}}$ with $\{0, 1, 2, 3, 5, 10\}$ steps, obtaining AUC-PRC values of 83.90, 95.98, 97.31, 94.96, 97.26, and 97.90, respectively. Even with a single step, GDPD achieves a substantial gain over the Base model (83.90), and only a few steps are sufficient to reach near-optimal performance; hence, we set NFEs to 5 in our experiments.

### A.3.2 COMPUTATIONAL COST ANALYSIS OF GDPD

We evaluate the computational overhead of GDPD by quantifying the training cost on the StarLightCurves dataset for LSTM3-100 $\to$ LSTM3-100 under the partialness level $e = 0.5L$.

Table 14: Training, memory, and inference cost comparison across distillation methods for LSTM3-100 $\to$ LSTM3-100.

| Method | Student Params (M) | Additional Params (M) | Total Train (h) | Epoch Time (s) | Step Time (ms) | GPU Mem (GB) | Inference (ms) |
|---|---|---|---|---|---|---|---|
| Base | 0.20 | 0 | 0.10 | 0.61 | 46.71 | 0.17 | 0.02 |
| Base KD | 0.20 | 0 | 0.10 | 0.61 | 46.88 | 0.17 | 0.02 |
| Fits | 0.20 | 0.01 | 0.10 | 0.62 | 48.34 | 0.22 | 0.02 |
| VID | 0.20 | 0.03 | 0.11 | 0.68 | 52.00 | 0.25 | 0.02 |
| DKD | 0.20 | 0 | 0.10 | 0.62 | 47.98 | 0.17 | 0.02 |
| Attention | 0.20 | 0 | 0.11 | 0.64 | 49.00 | 0.17 | 0.02 |
| RKD | 0.20 | 0 | 0.13 | 0.80 | 61.61 | 1.03 | 0.02 |
| GDPD | 0.20 | 0.21 | 0.14 | 0.85 | 65.14 | 0.25 | 0.02 |
| GDPD warm-up | 0.20 | 0.21 | 0.07 | 0.81 | 62.46 | 0.24 | 0.02 |
| GDPD guided phase | 0.20 | 0.21 | 0.08 | 0.89 | 69.21 | 0.25 | 0.02 |

Table 14 summarizes the results obtained using a single RTX-A6000/3090–class GPU. Below, we discuss the computational cost of each phase of GDPD.

**Teacher Training.** Training the teacher is identical to any standard KD pipeline and introduces no additional cost in GDPD.

**Diffusion-prior Training.** Trained in feature space, where the dimensionality is far lower than in the input domain, GDPD's diffusion training becomes significantly cheaper computationally. The diffusion prior is lightweight (only 0.206M trainable parameters, including the noise adapter) and is trained during the student's warm-up phase. The warm-up stage costs 0.81 s/epoch, compared to 0.61 s/epoch for the Base student. With a warm-up duration of 300 epochs, this adds only ∼1 minute of extra training in the evaluated setting. In practice, diffusion-prior training is comparable to training one additional Base classifier and remains far cheaper than ensemble-teacher distillation, while still providing the benefit of knowledge diversity.

**Diffusion-guided Training.** During GDPD's main distillation phase, posterior sampling uses only 5 NFEs, and all sampling is performed in feature space, making each reverse-diffusion step extremely cheap. The diffusion-guided stage costs 0.89 s/epoch, compared to 0.61 s/epoch for the Base student, an overhead of only ∼0.28 s per epoch. Over 300 epochs, this amounts to approximately 1.2 minutes of additional training time on the evaluated dataset.

**Overall Training Cost.** In the evaluated setting, training the Base and Base-KD requires 0.10 h, whereas GDPD (warm-up + guided phase) requires 0.14 h, adding only ∼2.4 minutes of extra training. The overall training cost of GDPD is comparable to widely adopted feature-distillation methods such as RKD and VID, while delivering substantially higher performance (Table 2). GDPD also maintains a low memory footprint of 0.24-0.25 GB, similar to Fits and VID, only slightly above logits-based KD (0.17 GB), and far below memory-intensive methods such as RKD (1.03 GB). Despite incorporating a diffusion prior, GDPD introduces minimal memory overhead. This modest training overhead is well justified by the consistent and significant performance improvements over conventional KD baselines and the Base classifier (Table 2).

**Inference Cost.** Inference cost is unchanged: the GDPD achieves 0.02 ms/sample, identical to the Base. All additional computation occurs only during training, while the deployed model remains as efficient as the Base classifier.

Table 15: Effect of inference diffusion steps on training cost, memory and inference time.

| Inference Steps | Total Train (h) | Epoch Time (s) | Step Time (ms) | GPU Mem (GB) | Inference (ms) | AUC-PRC (%) |
|---|---|---|---|---|---|---|
| 0 | 0.10 | 0.61 | 46.71 | 0.17 | 0.02 | 83.90 |
| 1 | 0.13 | 0.77 | 59.03 | 0.24 | 0.02 | 95.98 |
| 2 | 0.13 | 0.80 | 61.86 | 0.24 | 0.02 | 97.31 |
| 3 | 0.13 | 0.81 | 62.45 | 0.25 | 0.02 | 94.96 |
| 5 | 0.14 | 0.85 | 65.14 | 0.25 | 0.02 | 97.26 |
| 10 | 0.16 | 0.95 | 72.79 | 0.25 | 0.02 | 97.90 |

**Training Cost vs. Inference Steps.** Table 15 reports how the computational cost varies with the number of inference steps (equivalently, the NFEs in our implementation). Increasing inference steps slightly raises per-epoch time and memory. From 1 to 5 steps (our default), epoch time increases from 0.77 s to 0.85 s and memory from 0.24 GB to 0.25 GB. Even at 10 steps, memory remains at 0.25 GB and total training stays below 0.16 h. We adopt 5 steps as the best cost-performance trade-off.

**Training Cost vs. Posterior Samples.** Table 16 summarizes how training cost scales with the number of posterior samples. The cost grows roughly linearly, as each sample requires an additional draw. Epoch time rises from 0.85 s (1 sample) to 1.44 s (5 samples), and total training from 0.14 h to 0.24 h, while GPU memory remains fixed at 0.25 GB. We adopt a single sample as the best cost-performance trade-off.

Table 16: Effect of the number of posterior samples on training cost, memory and inference time.

| Posterior Samples (J) | Total Train (h) | Epoch Time (s) | Step Time (ms) | GPU Mem (GB) | Inference (ms) | AUC-PRC (%) |
|---|---|---|---|---|---|---|
| 0 | 0.10 | 0.61 | 46.71 | 0.17 | 0.02 | 83.90 |
| 1 | 0.14 | 0.85 | 65.14 | 0.25 | 0.02 | 97.26 |
| 2 | 0.17 | 0.99 | 76.05 | 0.25 | 0.02 | 97.34 |
| 3 | 0.19 | 1.11 | 85.99 | 0.25 | 0.02 | 97.13 |
| 4 | 0.22 | 1.30 | 100.17 | 0.25 | 0.02 | 97.87 |
| 5 | 0.24 | 1.44 | 111.07 | 0.25 | 0.02 | 97.78 |

Both ablations show that GDPD's diffusion controls offer a flexible cost-performance trade-off with minimal memory overhead and no effect on inference speed.

### A.3.3 EFFECT OF TEACHER–STUDENT LAYER SELECTION.

To assess the impact of layer choice, we perform a layer-wise ablation by applying GDPD across all teacher-student layer combinations in the LSTM3-100 $\rightarrow$ LSTM3-100 under the partialness level $e = 0.5L$ (Table 17). We additionally evaluate a "1+2+3" variant in which the diffusion prior is trained on features from all three teacher layers. Across these ten configurations, we compute the average rank and mean AUC-PRC over eight UCR datasets. The final-layer distillation ($3{\rightarrow}3$) reports the best performance, while distillation involving shallow layers (1 or 2) performs slightly lower yet remains close. The multi-layer prior (1+2+3) performs comparably but no better than final-layer distillation, suggesting that deep teacher features alone are the most effective.

We further report layer-wise averages and rank calculations for both teacher and student, and the results indicate that the final layer is the strongest choice for both. Accordingly, all our experiments use the final-layer features.

Table 17: Cross-layer GDPD performance for LSTM3-100 $\rightarrow$ LSTM3-100 on eight UCR datasets. Each cell reports **Avg. Rank $\downarrow$ / Avg. AUC-PRC $\uparrow$**. Layer-wise averages and rank calculations are also provided for each layer. "1+2+3" denotes a diffusion prior trained on feature representations from all three teacher layers.

| Teacher Layer | Student Layer | | | Teacher Layer Avg. |
|---|---|---|---|---|
| | 1 | 2 | 3 | |
| 1 | 8.88 / 81.35 | 6.63 / 82.02 | 5.50 / 82.13 | 2.42 / 81.83 |
| 2 | 6.00 / 82.04 | 5.25 / 82.84 | 4.00 / 83.28 | 1.83 / 82.72 |
| 3 | 5.38 / 82.25 | 4.13 / 82.92 | **3.89 / 84.03** | **1.75 / 83.07** |
| 1+2+3 | – | – | 5.38 / 81.56 | – |
| **Student Layer Avg.** | 2.38 / 80.88 | 1.88 / 82.59 | **1.75 / 83.15** | |

### A.3.4 GDPD IS ROBUST ACROSS DIFFERENT NETWORK ARCHITECTURES.

In Table 18, we report results for two distillation settings: 1) similar teacher–student architectures (Inception55-32 $\rightarrow$ Inception55-32, Resnet32-64 $\rightarrow$ Resnet32-64) and 2) cross architectures (Inception55-32 $\rightarrow$ Resnet32-64, Resnet32-64 $\rightarrow$ Inception55-32). Across both settings, GDPD consistently achieves the highest performance gains, measured by average AUC-PRC, lowest average rank, and the greatest number of top-1 wins, demonstrating its effectiveness for both similar- and cross-architecture distillation.

Table 18: Summary of similar- and cross-architecture distillation results on 12 UCR datasets. Each cell reports Avg. AUC-PRC / Avg. Rank / Num. Top-1 wins.

| | Inception $\rightarrow$ Inception | Inception $\rightarrow$ ResNet | ResNet $\rightarrow$ ResNet | ResNet $\rightarrow$ Inception |
|---|---|---|---|---|
| Base | 87.71 / 2.92 / 0 | 81.47 / 2.83 / 2 | 88.72 / 3.33 / 0 | 71.61 / 3.58 / 0 |
| Base-KD | 88.26 / 3.25 / 0 | 84.66 / 2.50 / 1 | 89.37 / 2.33 / 2 | 75.05 / 2.42 / 1 |
| Fits | 88.98 / 2.58 / 1 | 82.47 / 3.33 / 0 | 89.26 / 2.42 / 2 | 77.06 / 2.75 / 1 |
| GDPD | **92.29 / 1.17 / 11** | **87.79 / 1.17 / 10** | **90.85 / 1.58 / 8** | **81.05 / 1.08 / 11** |

A.3.5 GDPD DEGRADES GRACEFULLY UNDER WEAK TEACHER SUPERVISION.

To assess robustness when the teacher provides poorly structured feature spaces, we construct a sequence of increasingly degraded teachers on the StarLightCurves dataset by reducing the amount of training data and injecting label noise. We train four weak teachers: (WT-1) 0% training data reduction and 0% label noise, (WT-2) 25% training data reduction and 0% label noise, (WT-3) 25% training data reduction and 10% label noise, and (WT-4) 50% training data reduction and 25% label noise. Figure 3 reports how each baseline responds to these progressively degraded supervision for Inception55-32 $\rightarrow$ Inception55-32 under the partialness level $e = 0.5L$. Across the four weak-teacher settings, all methods degrade as supervision quality decreases, but GDPD consistently maintains the highest performance and shows the slowest rate of decline.

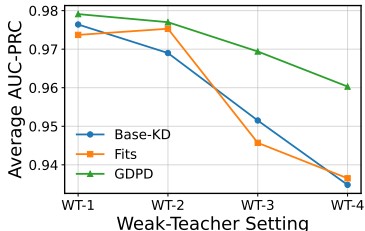

Figure 3: Performance comparison under four increasingly degraded teacher configurations (WT-1 to WT-4), showing how each method responds to weak-teacher supervision.

A.3.6 GDPD IS ROBUST ACROSS DIFFERENT EARLINESS LEVELS.

Table 19 reports results for earliness levels $0.5L$ and $L$, which are complementary to the main results presented in Table 1.

Table 19: Summary of performance at earliness levels 0.5L and L (full length) on 12 UCR datasets. Best values are in **bold**. Rows marked with $\downarrow$ indicate lower-is-better. GDPD achieves the highest AUC-PRC, the lowest rank, and the largest number of Top-1 wins.

| | Earliness=0.5L | | | | | Earliness=L (full) | | | |
|---|---|---|---|---|---|---|---|---|---|
| | Base | Base-KD | Fits | GDPD | | Base | Base-KD | Fits | GDPD |
| Avg. AUC-PRC | 71.60 | 78.14 | 77.28 | **84.64** | | 87.32 | 89.29 | 88.73 | **91.21** |
| Avg. Rank $\downarrow$ | 3.50 | 2.33 | 2.92 | **1.17** | | 3.33 | 2.08 | 2.75 | **1.58** |
| Num. Top-1 | 0 | 2 | 0 | **10** | | 0 | 5 | 3 | **6** |
| Wins/Draws | 12 | 10 | 12 | – | | 11 | 8 | 10 | – |
| Losses $\downarrow$ | 0 | 2 | 0 | – | | 1 | 4 | 2 | – |

A.3.7 FULL RESULTS FOR SUMMARIES REPORTED IN THE MAIN TEXT

**Robustness to Different Earliness Levels.** Table 20 provides the complete results of evaluating GDPD under different earliness levels on 12 UCR datasets.

**Comparison with Different Distillation Objectives.** The full results comparing GDPD with existing KD variants, evaluated at earliness $e = 0.5L$, are provided in Table 21.

**Time-wise and Channel-wise Partialness for Multivariate Datasets.** The detailed results for time-wise partialness (evaluated at $e = 0.5L$) and time+channel-wise partialness (evaluated at $e = 0.5L, m = 0.5M$) in the case of LSTM3-100 $\rightarrow$ LSTM3-100 distillation are provided in Table 22. The corresponding full results for Inception55-32 $\rightarrow$ Inception55-32 distillation under the same earliness settings are reported in Table 23.

**GDPD in Model Compression.** The complete results for two compression targets under two earliness levels are presented in Table 24.

**GDPD in Self-distillation.** The detailed self-distillation results (with similar model capacity and without earliness) are summarized in Table 25.

Table 20: Detailed UCR results at earliness $e \in \{0.2L, 0.4L, 0.5L, 0.6L, 0.8L, L\}$ for **LSTM** → **LSTM**. Best per row in **bold**.

(a) $e = 0.2L$

| Dataset | Base | Base-KD | Fits | GDPD |
|---|---|---|---|---|
| CBF | 64.08 | **69.24** | 68.44 | 68.96 |
| Coffee | 90.76 | 98.88 | 91.07 | **99.34** |
| ECG200 | 66.08 | 76.88 | 79.81 | **80.19** |
| ECGFiveDays | 56.78 | 57.19 | 56.32 | **57.70** |
| Gun_Point | 77.38 | 75.18 | 81.38 | **84.80** |
| FaceAll | 31.90 | 48.29 | 41.88 | **52.84** |
| ItalyPowerDemand | 67.45 | 69.34 | 66.36 | **73.30** |
| NonInvasiveFatal1 | 51.92 | 48.74 | 50.16 | **67.34** |
| StarLightCurves | 77.35 | 93.40 | 86.80 | **94.83** |
| synthetic_control | 81.02 | **87.57** | 83.20 | 87.52 |
| Trace | 39.74 | 51.43 | 40.31 | **54.94** |
| TwoLeadECG | 59.22 | 54.56 | 63.88 | **64.20** |
| Avg. AUC-PRC | 63.64 | 69.23 | 67.47 | **73.83** |
| Avg. Rank | 3.50 | 2.42 | 2.92 | **1.17** |
| Num. Top-1 | 0 | 2 | 0 | **10** |
| Wins/Draws | 12 | 10 | 12 | – |
| Losses | **0** | 2 | **0** | – |

(b) $e = 0.4L$

| Dataset | Base | Base-KD | Fits | GDPD |
|---|---|---|---|---|
| CBF | 93.87 | 89.50 | 88.11 | **94.35** |
| Coffee | 64.95 | 88.28 | 78.14 | **90.13** |
| ECG200 | 68.26 | 78.93 | 70.34 | **80.15** |
| ECGFiveDays | 55.40 | **56.38** | 56.30 | 55.82 |
| Gun_Point | 64.44 | 74.87 | 68.31 | **84.59** |
| FaceAll | 55.67 | 69.90 | 72.44 | **73.78** |
| ItalyPowerDemand | 79.74 | 79.43 | 79.93 | **81.84** |
| NonInvasiveFatal1 | 49.02 | 67.35 | **72.23** | 72.20 |
| StarLightCurves | 93.68 | 97.13 | 95.20 | **97.43** |
| synthetic_control | 82.32 | 81.66 | 86.59 | **90.87** |
| Trace | 68.82 | 73.42 | 71.78 | **74.05** |
| TwoLeadECG | 69.10 | 79.47 | 64.93 | **85.17** |
| Avg. AUC-PRC | **81.70** | 78.03 | 75.36 | **81.70** |
| Avg. Rank | 3.58 | 2.50 | 2.67 | **1.25** |
| Num. Top-1 | 0 | 1 | 1 | **10** |
| Wins/Draws | 12 | 11 | 10 | – |
| Losses | **0** | 1 | 2 | – |

(c) $e = 0.5L$

| Dataset | Base | Base-KD | Fits | GDPD |
|---|---|---|---|---|
| CBF | 90.29 | **97.39** | 95.04 | 95.44 |
| Coffee | 71.26 | 83.28 | 82.98 | **85.28** |
| ECG200 | 79.31 | **83.61** | 79.30 | 82.95 |
| ECGFiveDays | 48.33 | 61.44 | 66.72 | **73.85** |
| Gun_Point | 74.21 | 79.22 | 78.90 | **93.55** |
| FaceAll | 53.09 | 71.48 | 70.09 | **76.08** |
| ItalyPowerDemand | 82.53 | 87.26 | 82.53 | **91.71** |
| NonInvasiveFatal1 | 71.68 | 72.12 | 71.89 | **73.32** |
| StarLightCurves | 83.90 | 95.20 | 81.87 | **97.26** |
| synthetic_control | 63.82 | 61.46 | 63.08 | **82.94** |
| Trace | 63.59 | 63.16 | 72.52 | **74.87** |
| TwoLeadECG | 77.19 | 82.07 | 82.45 | **88.44** |
| Avg. AUC-PRC | 71.60 | 78.14 | 77.28 | **84.64** |
| Avg. Rank | 3.50 | 2.33 | 2.92 | **1.17** |
| Num. Top-1 | 0 | 2 | 0 | **10** |
| Wins/Draws | 12 | 10 | 12 | – |
| Losses | **0** | 2 | **0** | – |

(d) $e = 0.6L$

| Dataset | Base | Base-KD | Fits | GDPD |
|---|---|---|---|---|
| CBF | 82.61 | **90.68** | 82.76 | 89.88 |
| Coffee | 74.35 | 100 | 90.64 | 100 |
| ECG200 | 76.09 | 80.78 | **83.55** | 81.26 |
| ECGFiveDays | 78.08 | 86.82 | 81.46 | **89.90** |
| Gun_Point | 69.81 | 74.10 | **77.99** | 76.36 |
| FaceAll | 69.21 | 69.68 | 70.64 | **75.61** |
| ItalyPowerDemand | 93.04 | 92.46 | 93.02 | **95.32** |
| NonInvasiveFatal1 | 68.91 | 70.97 | 73.53 | **74.25** |
| StarLightCurves | 81.15 | 94.02 | 95.23 | **96.97** |
| synthetic_control | 99.69 | 99.64 | 99.86 | **99.87** |
| Trace | 50.18 | **67.41** | 55.64 | 65.34 |
| TwoLeadECG | 78.32 | 77.79 | 69.51 | **87.27** |
| Avg. AUC-PRC | 76.79 | 83.70 | 81.15 | **86.00** |
| Avg. Rank | 3.58 | 2.58 | 2.42 | **1.33** |
| Num. Top-1 | 0 | 3 | 2 | **8** |
| Wins/Draws | 12 | 10 | 10 | – |
| Losses | **0** | 2 | 2 | – |

(e) $e = 0.8L$

| Dataset | Base | Base-KD | Fits | GDPD |
|---|---|---|---|---|
| CBF | 93.37 | 93.44 | **95.46** | 95.26 |
| Coffee | 98.23 | 99.51 | 98.34 | **100.00** |
| ECG200 | 83.32 | 77.94 | 82.69 | **85.01** |
| ECGFiveDays | 59.33 | 74.01 | 66.29 | **79.09** |
| Gun_Point | 92.29 | 92.71 | 92.82 | **93.20** |
| FaceAll | 49.31 | 73.71 | 73.33 | **78.59** |
| ItalyPowerDemand | 92.90 | 96.37 | 94.17 | **98.07** |
| NonInvasiveFatal1 | 48.75 | 63.70 | 69.58 | **72.81** |
| StarLightCurves | 83.34 | 96.45 | 80.79 | **96.46** |
| synthetic_control | 98.09 | 99.11 | 99.05 | **99.33** |
| Trace | 48.52 | 60.82 | 54.29 | **75.38** |
| TwoLeadECG | 86.05 | 89.56 | 86.04 | **95.09** |
| Avg. AUC-PRC | 77.79 | 84.78 | 82.74 | **89.02** |
| Avg. Rank | 3.67 | 2.42 | 2.83 | **1.08** |
| Num. Top-1 | 0 | 0 | 1 | **11** |
| Wins/Draws | 12 | 12 | 11 | – |
| Losses | **0** | **0** | 1 | – |

(f) $e = L$

| Dataset | Base | Base-KD | Fits | GDPD |
|---|---|---|---|---|
| CBF | 95.08 | **99.50** | 91.49 | 99.26 |
| Coffee | 99.67 | 99.67 | **100.00** | **100.00** |
| ECG200 | 78.75 | **80.36** | 77.30 | 79.18 |
| ECGFiveDays | 92.62 | 90.52 | 85.44 | **95.37** |
| Gun_Point | 96.11 | 92.20 | **96.54** | 94.11 |
| FaceAll | 78.83 | 83.53 | 82.84 | **85.94** |
| ItalyPowerDemand | 98.68 | **99.21** | 98.55 | **99.21** |
| NonInvasiveFatal1 | 84.49 | **88.38** | 85.44 | 88.35 |
| StarLightCurves | 96.12 | 97.19 | 96.92 | **97.45** |
| synthetic_control | 99.08 | **99.73** | 99.59 | 99.66 |
| Trace | 60.81 | 72.30 | **77.77** | 77.26 |
| TwoLeadECG | 67.62 | 68.84 | 72.93 | **78.71** |
| Avg. AUC-PRC | 87.32 | 89.29 | 88.73 | **91.21** |
| Avg. Rank | 3.33 | 2.08 | 2.75 | **1.58** |
| Num. Top-1 | 0 | **5** | 3 | 6 |
| Wins/Draws | 11 | 8 | 10 | – |
| Losses | 1 | 4 | 2 | – |

Table 21: Performance across 12 UCR datasets for **LSTM** → **LSTM**. Best per row in **bold**; second best is underlined. (For *Avg. Rank*, lower is better.)

| Dataset | Base | Base-KD | Fits | VID | DKD | Attention | DT2W | RKD | RKD-A | PKT | TeKAP | TTM | GDPD |
|---|---|---|---|---|---|---|---|---|---|---|---|---|---|
| CBF | 90.29 | **97.39** | 95.04 | 84.18 | 90.90 | 80.20 | 46.60 | 86.67 | 80.12 | 75.95 | 79.70 | 86.41 | 95.44 |
| Coffee | 71.26 | 83.28 | 82.98 | 86.33 | 84.41 | 82.59 | 72.51 | 83.18 | **96.47** | 85.67 | 84.53 | 85.02 | 85.28 |
| ECG200 | 79.31 | **83.61** | 79.30 | 81.01 | 79.80 | 83.28 | 81.87 | 79.13 | 81.08 | 80.22 | 81.47 | 82.94 | 82.95 |
| ECGFiveDays | 48.33 | 61.44 | 66.72 | 70.66 | 59.67 | 66.97 | 51.13 | **82.85** | 65.96 | 69.77 | 63.16 | 71.95 | 73.85 |
| Gun_Point | 74.21 | 79.22 | 78.90 | 76.42 | 63.98 | 65.99 | 56.31 | 84.58 | 70.85 | 64.36 | 68.55 | 74.09 | **93.55** |
| FaceAll | 53.09 | 71.48 | 70.09 | 72.73 | 62.24 | 71.08 | 13.97 | 70.63 | 68.93 | 73.92 | 74.90 | 68.05 | **76.08** |
| ItalyPowerDemand | 82.53 | 87.26 | 82.53 | 88.94 | 84.03 | 84.77 | 90.61 | **94.77** | 80.13 | 88.07 | 88.87 | 91.71 |
| NonInvasiveFatalECG1 | 71.68 | 72.12 | 71.89 | 55.39 | 51.22 | 61.79 | 16.61 | 59.14 | 57.48 | 63.45 | 65.73 | 62.30 | **73.32** |
| StarLightCurves | 83.90 | 95.20 | 81.87 | 89.81 | 89.24 | 87.65 | 72.41 | 92.47 | 96.46 | 93.44 | 91.87 | 87.57 | **97.26** |
| synthetic_control | 63.82 | 61.46 | 63.08 | 74.58 | 65.13 | 67.73 | 48.87 | **98.41** | 96.76 | 78.07 | 63.13 | 70.39 | 82.94 |
| Trace | 63.59 | 63.16 | 72.52 | 64.28 | 75.44 | 69.26 | 55.01 | 64.35 | 65.75 | 77.23 | **78.31** | 76.92 | 74.87 |
| TwoLeadECG | 77.19 | 82.07 | 82.45 | 81.93 | 85.37 | 84.37 | 57.95 | 82.99 | 76.53 | 80.10 | 79.05 | 82.05 | **88.44** |
| Avg. AUC-PRC | 71.60 | 78.14 | 77.28 | 77.19 | 74.29 | 75.47 | 55.32 | 79.89 | 79.26 | 76.86 | 76.54 | 78.05 | **84.64** |
| Test-Fidelity | 66.75 | 72.13 | 69.39 | 71.32 | 69.36 | 69.20 | 52.59 | 74.34 | 71.22 | 71.05 | 69.53 | 70.24 | **77.58** |
| Avg. Rank ↓ | 9.67 | 5.83 | 7.58 | 6.33 | 8.25 | 7.25 | 11.42 | 6.33 | 6.42 | 6.67 | 6.75 | 6.17 | **2.25** |
| Num. Top-1 | 0 | 2 | 0 | 0 | 0 | 0 | 0 | 2 | 2 | 0 | 1 | 0 | **5** |
| Num. Top-3 | 0 | 5 | 2 | 1 | 1 | 2 | 0 | 4 | 4 | 3 | 2 | 2 | **10** |
| Wins/Draws | 12 | 10 | 12 | 11 | 11 | 11 | 12 | 9 | 9 | 10 | 11 | 11 | – |
| Losses | 0 | 1 | 0 | 1 | 1 | 1 | 0 | 3 | 3 | 2 | 1 | 1 | – |

Table 22: Detailed results for **LSTM** $\rightarrow$ **LSTM** on multivariate datasets under Time-wise partialness ($e = 0.5L$) and Time+Channel partialness ($e = 0.5L, m = 0.5M$). Best result per row in **bold**.

(a) Time-wise partialness

| Dataset | Base | Base-KD | Fits | GDPD |
|---|---|---|---|---|
| ArticularyWordRecog. | 87.55 | **90.26** | 89.66 | 89.88 |
| BasicMotions | 93.54 | 91.89 | 92.67 | **97.33** |
| Cricket | 83.39 | 85.27 | 75.77 | **87.95** |
| ERing | 64.25 | 62.69 | 66.74 | **67.55** |
| JapaneseVowels | 94.39 | 96.49 | 95.98 | **96.99** |
| Libras | 49.49 | 65.91 | 64.70 | **66.94** |
| NATOPS | 80.14 | 81.08 | 82.31 | **84.93** |
| PEMS-SF | 71.70 | **76.35** | 73.79 | 75.50 |
| PenDigits | 95.95 | **96.14** | 95.99 | 95.14 |
| RacketSports | 82.36 | 85.20 | **87.82** | 86.64 |
| SelfRegulationSCP1 | 84.28 | 83.24 | 85.89 | **86.88** |
| UWaveGestureLibrary | 65.82 | 68.98 | **72.08** | 70.73 |
| Avg. AUC-PRC | 79.41 | 81.96 | 81.95 | **83.87** |
| Avg. Rank | 3.50 | 2.50 | 2.42 | **1.58** |
| Num. Top-1 | 0 | 3 | 2 | **7** |
| Wins/Draws | 11 | 9 | 9 | – |
| Losses | 1 | 3 | 3 | – |

(b) Time+Channel partialness

| Dataset | Base | Base-KD | Fits | GDPD |
|---|---|---|---|---|
| ArticularyWordRecog. | 60.41 | 64.97 | **69.41** | 66.57 |
| BasicMotions | 85.18 | 94.85 | 94.31 | **97.44** |
| Cricket | 66.27 | 67.46 | 70.12 | **71.69** |
| ERing | 46.52 | 52.98 | 46.42 | **55.23** |
| JapaneseVowels | 92.26 | 91.70 | 92.53 | **93.04** |
| Libras | 35.16 | 37.35 | 38.32 | **41.27** |
| NATOPS | 86.86 | 85.27 | 86.86 | **87.64** |
| PEMS-SF | 66.98 | 69.60 | 69.17 | **73.57** |
| PenDigits | 75.46 | **75.61** | 75.55 | 70.91 |
| RacketSports | 74.32 | 74.45 | 77.35 | **79.18** |
| SelfRegulationSCP1 | 84.23 | 87.99 | 83.97 | **91.11** |
| UWaveGestureLibrary | 51.37 | 51.12 | **56.29** | 55.56 |
| Avg. AUC-PRC | 68.75 | 71.11 | 71.69 | **73.60** |
| Avg. Rank | 3.42 | 2.75 | 2.33 | **1.42** |
| Num. Top-1 | 0 | 1 | 2 | **9** |
| Wins/Draws | 11 | 11 | 9 | – |
| Losses | 1 | 1 | 3 | – |

Table 23: Detailed results for **Inception** $\rightarrow$ **Inception** on multivariate datasets under Time-wise partialness ($e = 0.5L$) and Time+Channel partialness ($e = 0.5L, m = 0.5M$). Best result per row in **bold**.

(a) Time-wise partialness

| Dataset | Base | Base-KD | Fits | GDPD |
|---|---|---|---|---|
| ArticularyWordRecog. | 92.23 | 96.32 | 84.62 | **96.92** |
| BasicMotions | 99.78 | **100** | **100** | **100** |
| Cricket | 97.75 | **98.96** | 97.03 | 98.69 |
| ERing | 76.65 | 79.26 | 70.15 | **80.21** |
| JapaneseVowels | 98.43 | 99.04 | 96.26 | **99.19** |
| Libras | 68.78 | 81.28 | 81.20 | **82.70** |
| NATOPS | 86.94 | 84.78 | 82.01 | **89.65** |
| PEMS-SF | 57.84 | 65.07 | 65.45 | **77.70** |
| PenDigits | 90.68 | 95.37 | **95.39** | 94.69 |
| RacketSports | 88.84 | 86.50 | 85.53 | **88.91** |
| SelfRegulationSCP1 | 95.13 | **96.96** | 95.71 | 96.91 |
| UWaveGestureLibrary | 81.55 | 81.66 | 71.49 | **84.27** |
| Avg. AUC-PRC | 86.22 | 88.77 | 85.40 | **90.82** |
| Avg. Rank | 3.25 | 2.00 | 3.17 | **1.33** |
| Num. Top-1 | 0 | 3 | 2 | **9** |
| Wins/Draws | 12 | 9 | 11 | – |
| Losses | **0** | 3 | 1 | – |

(b) Time+Channel partialness

| Dataset | Base | Base-KD | Fits | GDPD |
|---|---|---|---|---|
| ArticularyWordRecog. | 69.44 | 76.01 | 68.16 | **79.56** |
| BasicMotions | 96.78 | **100** | **100** | **100** |
| Cricket | 84.03 | **87.61** | 81.96 | 86.66 |
| ERing | 71.87 | 71.39 | 70.40 | **72.42** |
| JapaneseVowels | 94.95 | 96.40 | 93.95 | **96.65** |
| Libras | 39.60 | 39.31 | 40.96 | **45.05** |
| NATOPS | 87.48 | 88.51 | 87.20 | **90.35** |
| PEMS-SF | 63.80 | 64.70 | 61.31 | **75.71** |
| PenDigits | 75.05 | **75.49** | 74.73 | 71.90 |
| RacketSports | 82.15 | 82.35 | 75.82 | **83.73** |
| SelfRegulationSCP1 | 95.32 | 96.45 | 95.70 | **96.85** |
| UWaveGestureLibrary | 58.43 | 56.87 | 53.91 | **64.05** |
| Avg. AUC-PRC | 76.58 | 77.92 | 75.34 | **80.24** |
| Avg. Rank | 2.92 | 2.08 | 3.42 | **1.33** |
| Num. Top-1 | 0 | 3 | 1 | **10** |
| Wins/Draws | 11 | 10 | 11 | – |
| Losses | 1 | 2 | 1 | – |

## A.4 DISCUSSION

### A.4.1 CONDITIONING STRATEGY IN GDPD

GDPD operates in an inverse diffusion setting, where the guidance is provided by the measurement or partial observation. The conditioning mechanism in inverse diffusion is fundamentally different from classifier- or classifier-free guidance used in guided generation. Below, we justify our use of simple initialization-based conditioning in GDPD by outlining why other common conditioning strategies in the inverse-diffusion are unsuitable for our setting.

**Conditional Denoisers.** A common approach is to feed the measurement ($z_{\text{short}}$) into the denoiser during both training and sampling, allowing the denoiser to learn how to influence the output $z_{\text{long}}$. This resembles classifier-free guidance in guided generation. However, this is incompatible with GDPD since conditioning signals are non-stationary. During training, the student keeps learning, and its $z_{\text{short}}$ features change continuously. A conditional denoiser trained on earlier student states quickly becomes invalid, because it no longer reflects the student's updated representation. Keeping it aligned would require retraining the denoiser at every student update. For this reason, GDPD cannot rely on conditional denoisers.

**Projection or Data-consistency Updates.** These methods apply an unconditional denoiser and then project the output onto the set of measurements to enforce consistency with the observation,

Table 24: Complete model compression results under two earliness levels ($e = 0.5L, L$) and two compression targets (LSTM3-100→LSTM1-8, LSTM3-100→LSTM2-32). Best per row is in **bold**.

| Earliness | Dataset | LSTM3-100 → LSTM1-8 | | | | LSTM3-100 → LSTM2-32 | | | |
|---|---|---|---|---|---|---|---|---|---|
| | | Base | Base-KD | Fits | GDPD | Base | Base-KD | Fits | GDPD |
| $e = 0.5L$ | CBF | 58.95 | 60.45 | 58.95 | **70.17** | 90.40 | 90.58 | 90.49 | **92.61** |
| | Coffee | 71.12 | 69.93 | 71.45 | **78.00** | 80.29 | **86.16** | 77.06 | 84.73 |
| | ECG200 | 83.51 | **84.94** | 84.76 | 84.06 | 81.54 | 79.39 | 79.79 | **82.28** |
| | ECGFiveDays | 63.31 | 73.51 | 66.44 | **77.41** | 74.47 | 63.53 | 74.23 | **75.10** |
| | Gun_Point | 60.93 | 66.56 | 61.06 | **75.27** | 62.93 | 78.14 | 76.75 | **80.19** |
| | FaceAll | 26.84 | 28.81 | 26.84 | **47.19** | 53.96 | 62.95 | 68.04 | **74.60** |
| | ItalyPowerDemand | 85.00 | 86.41 | 85.00 | **93.35** | 92.10 | **92.76** | 90.27 | 92.37 |
| | NonInvasiveFatal1 | 18.20 | 19.51 | 18.24 | **33.03** | 50.63 | 36.91 | 57.88 | **76.27** |
| | StarLightCurves | 75.63 | 77.66 | 76.11 | **80.33** | 77.18 | 83.13 | 93.92 | **94.22** |
| | synthetic_control | 53.78 | 56.14 | 63.52 | **74.62** | 61.60 | 68.65 | 76.42 | **93.73** |
| | Trace | 49.02 | 48.81 | 49.02 | **73.08** | 67.17 | 69.78 | 62.10 | **73.23** |
| | TwoLeadECG | 56.67 | 70.90 | 56.67 | **86.60** | 83.50 | 87.10 | **92.21** | 84.68 |
| | Avg. AUC-PRC | 58.58 | 61.97 | 59.84 | **72.76** | 72.98 | 74.92 | 78.26 | **83.67** |
| | Avg. Rank | 3.42 | 2.33 | 2.67 | **1.17** | 3.33 | 2.58 | 2.75 | **1.33** |
| | Num. Top-1 | 0 | 1 | 0 | **11** | 0 | 2 | 1 | **9** |
| | Wins/Draws | 12 | 11 | 11 | – | 12 | 9 | 11 | – |
| | Losses | 0 | 1 | 1 | – | 0 | 3 | 1 | – |
| $e = L$ | CBF | 65.44 | 61.32 | 64.41 | **65.89** | 77.17 | 79.80 | 78.50 | **86.78** |
| | Coffee | 87.08 | 87.79 | **97.38** | 91.64 | 99.51 | **100** | 99.67 | **100** |
| | ECG200 | 70.71 | 70.67 | 70.92 | **74.09** | 72.68 | 73.33 | 76.15 | **80.45** |
| | ECGFiveDays | 75.76 | **88.23** | 79.52 | 87.88 | 72.22 | 91.97 | 69.00 | **93.22** |
| | Gun_Point | 94.75 | 96.32 | 95.63 | **96.77** | 91.40 | **98.47** | 94.95 | 98.00 |
| | FaceAll | 42.91 | 45.82 | 43.81 | **47.31** | 76.09 | 81.98 | 80.12 | **82.59** |
| | ItalyPowerDemand | 98.46 | **99.03** | 98.46 | 98.99 | 98.18 | 98.67 | 98.24 | **98.68** |
| | NonInvasiveFatal1 | 43.81 | 44.53 | 43.45 | **44.78** | 86.07 | **88.88** | 86.47 | 88.40 |
| | StarLightCurves | 93.20 | 92.62 | **93.65** | 93.27 | 96.40 | **97.24** | 96.60 | 97.20 |
| | synthetic_control | 87.73 | 96.99 | 93.36 | **97.83** | 99.50 | 99.38 | 99.44 | **99.54** |
| | Trace | 69.89 | 75.10 | 69.64 | **75.44** | 69.52 | 75.58 | **78.00** | 78.00 |
| | TwoLeadECG | 69.25 | **72.99** | 67.62 | 72.03 | 73.28 | 72.02 | 74.03 | **75.25** |
| | Avg. AUC-PRC | 74.92 | 77.62 | 76.49 | **78.83** | 84.34 | 88.11 | 85.93 | **89.84** |
| | Avg. Rank | 3.33 | 2.33 | 2.83 | **1.42** | 3.67 | 2.17 | 2.75 | **1.25** |
| | Num. Top-1 | 0 | 3 | 2 | **7** | 0 | 4 | 1 | **9** |
| | Wins/Draws | 12 | 9 | 10 | – | 12 | 9 | 12 | – |
| | Losses | 0 | 3 | 2 | – | 0 | 3 | 0 | – |

Table 25: Detailed self-distillation results on 12 UCR datasets under three architectures. Best per row is in **bold**.

| Dataset | LSTM → LSTM | | | | Inception → Inception | | | | ResNet → ResNet | | | |
|---|---|---|---|---|---|---|---|---|---|---|---|---|
| | Teacher | Base-KD | Fits | GDPD | Teacher | Base-KD | Fits | GDPD | Teacher | Base-KD | Fits | GDPD |
| CBF | 95.08 | **99.50** | 91.49 | 99.26 | 99.19 | 99.91 | 96.51 | **99.94** | 99.72 | 99.69 | 99.61 | **99.76** |
| Coffee | 99.67 | 99.67 | **100** | **100** | 99.75 | **100** | 94.70 | **100** | **100** | **100** | **100** | **100** |
| ECG200 | 78.75 | **80.36** | 77.30 | 79.18 | 91.16 | **92.21** | 91.14 | 92.16 | 92.19 | 92.91 | **93.37** | 93.09 |
| ECGFiveDays | 92.62 | 90.52 | 85.44 | **95.37** | 95.94 | **100** | 94.51 | **100** | 98.25 | 97.29 | 97.84 | 97.63 |
| Gun_Point | 96.11 | 92.20 | **96.54** | 94.11 | 99.71 | **100** | 99.72 | 99.99 | **100** | **100** | **100** | **100** |
| FaceAll | 78.83 | 83.53 | 82.84 | **85.94** | 86.72 | 89.49 | 71.69 | **90.87** | 96.38 | 97.19 | 96.80 | **97.25** |
| ItalyPowerDemand | 98.68 | **99.21** | 98.55 | **99.21** | 98.90 | **99.05** | 98.01 | **99.05** | 98.69 | 98.99 | 98.96 | **99.00** |
| NonInvasiveFatalECG1 | 84.49 | **88.38** | 85.44 | 88.35 | 94.85 | 95.01 | 84.53 | **95.33** | 97.45 | 97.44 | 97.41 | **97.48** |
| StarLightCurves | 96.12 | 97.19 | 96.92 | **97.45** | 97.79 | 98.09 | 97.68 | **98.38** | 98.47 | 98.72 | 98.69 | **98.75** |
| synthetic_control | 99.08 | **99.73** | 99.59 | 99.66 | 99.97 | 99.95 | 99.29 | **100** | 99.98 | **99.99** | 99.94 | **99.99** |
| Trace | 60.81 | 72.30 | **77.77** | 77.26 | **100** | **100** | 99.54 | **100** | **100** | **100** | **100** | **100** |
| TwoLeadECG | 67.62 | 68.84 | 72.93 | **78.71** | 99.55 | **99.92** | 99.42 | 99.89 | **100** | 99.99 | **100** | **100** |
| Avg. AUC-PRC | 87.32 | 89.29 | 88.73 | **91.21** | 96.96 | 97.80 | 93.90 | **97.97** | 98.43 | 98.52 | 98.55 | **98.58** |
| Avg. Rank | 3.33 | 2.08 | 2.75 | **1.58** | 2.83 | 1.50 | 3.92 | **1.25** | 2.33 | 2.25 | 2.33 | **1.25** |
| Num. Top-1 | 0 | 5 | 3 | **6** | 1 | 7 | 0 | **9** | 5 | 4 | 5 | **10** |
| Wins/Draws | 11 | 8 | 10 | – | 12 | 9 | 12 | – | 11 | 12 | 10 | – |
| Losses | 1 | 4 | 2 | – | 0 | 3 | 0 | – | 1 | 0 | 2 | – |

typically assuming a known, fixed, and linear degradation model. In GDPD, the "degradation" from $z_{\text{long}}$ to $z_{\text{short}}$ is neither known, fixed, nor linear. The student features $z_{\text{short}}$ evolve throughout training, making any strict projection-based constraint incompatible with their non-stationary nature.

**Bayesian Likelihood-gradient Conditioning.** These methods use an unconditional denoiser and modify the score at each diffusion step with the gradient of the likelihood—i.e., how likely the current sample would produce the observed measurement. Because this likelihood is intractable, inverse-diffusion methods approximate it using assumptions tied to their specific degradation setting

(linear, nonlinear, noisy, etc.). This requires explicit knowledge of the forward degradation and introduces significant computational overhead. In GDPD, however, the degradation from $z_{long}$ to $z_{short}$ is unknown, dynamic, and not tied to any fixed physical process. Because the GDPD loss already supervises the outputs and the conditioning signal itself evolves during training, we treat this mapping as general and do not enforce strict measurement consistency at every diffusion step. Imposing such exact reconstruction constraints would be both unnecessary and computationally costly.

**Initialization-based Conditioning.** We adopt initialization-based conditioning because it accommodates the evolving nature of the student features (with the appropriate noise level) without requiring any retraining or modification of the conditioning mechanism. As the student updates, its $z_{short}$ features directly influence the diffusion trajectory, ensuring that the conditioning always reflects the student's current knowledge. Although this approach does not enforce strict measurement consistency, this is acceptable: the GDPD loss already supervises the reconstructed features, and the conditioning signal only needs to steer the generation process rather than impose exact reconstruction constraints.

### A.4.2 Limitations and Future Work.

Modeling teacher knowledge through a generative prior opens up a broad space of potential distillation objectives to foster many desirable teacher properties. In this work, we instantiate only one such formulation, and exploring alternative distillation objectives grounded in the generative prior remains a promising direction for future research. The computational cost of GDPD is higher than simple logits-based distillation methods (e.g., Base-KD, DKD), but it is comparable to established feature-based distillation objectives such as VID and RKD.

**Applicability Beyond Time Series Classification.** This work establishes GDPD as a principled distillation framework for partial time-series classification, validated across standard benchmarks and a real-world case study, and accompanied by practical guidance on key hyperparameter controls. The current implementation supervises posterior reconstructions using a classification loss and is therefore instantiated for classification. However, modeling teacher knowledge as a generative prior provides a novel and broadly applicable perspective on knowledge distillation, leaving several promising extensions of GDPD for future work. First, GDPD may be applicable to forecasting or multimodal learning with missing modalities by substituting the classification-based posterior supervision with the relevant task-specific objective. Second, the proposed framework can, in principle, be applied to partial-input classification in domains beyond time series, such as short-text classification (e.g., headlines or snippets) (Zhu et al., 2024). In textual data, topic shifts and the discrete nature of token representations make short contexts more ambiguous than temporally continuous signals (Zhu et al., 2024). Thus, when extending GDPD to textual domains, feasible earliness levels must be carefully validated, and potential mismatches in teacher–student vocabularies (reflected in their embedding layers) should be addressed. These considerations represent promising directions for future exploration.

## The Use of Large Language Models (LLMs)

In preparing this manuscript, we used OpenAI's ChatGPT (GPT-4) as a writing assistance tool. Its role was limited to polishing language for improved clarity, grammar, and readability in certain sections of the paper and appendix. The model did not contribute to research ideation, methodological design, experimental execution, data analysis, or interpretation of results. All scientific content, technical contributions, and conclusions are solely the work of the authors. We take full responsibility for the accuracy and integrity of the paper.

