# OpenReview forum: "Generative Diffusion Prior Distillation for Long-Context Knowledge Transfer"
_ICLR.cc/2026/Conference — ICLR 2026 Poster_

### Official Review · Reviewer_VsYh · 2025-10-30

**Soundness:** 2
**Presentation:** 3
**Contribution:** 2
**Rating:** 4
**Confidence:** 3

**Summary:**

The paper proposes Generative Diffusion Prior Distillation (GDPD), a novel knowledge distillation framework that treats the teacher’s full-context representation as a generative prior and uses diffusion models to approximate the distribution of teacher features. During distillation, the student’s partial observations are regarded as degraded evidence, and posterior sampling from the diffusion prior produces plausible teacher feature reconstructions that serve as soft supervision. This approach effectively bridges the gap between full-context teachers and partial-context students in time-series or incomplete-modality settings. Extensive experiments on UCR, UEA, and PhysioNet datasets demonstrate consistent gains over standard KD and feature-based methods, accompanied by detailed ablation studies and reproducibility details.

**Strengths:**

1. The paper frame the teacher’s representation as a learnable generative prior, leveraging diffusion modeling to capture the intrinsic uncertainty and diversity of knowledge transfer. This view enriches the typically deterministic KD paradigm with a stochastic, distributional interpretation.
2. The experiments are extensive and systematically designed, with convincing improvements and well-controlled ablations.

**Weaknesses:**

1. While GDPD is conceptually elegant, the paper’s treatment of the diffusion prior remains largely phenomenological—its success is empirically shown, yet its mechanistic role in shaping the student’s feature space is underexplored. The work stops short of analyzing what kind of uncertainty the diffusion prior captures (epistemic vs. aleatoric), or how this aligns with the teacher’s representational manifold. Consequently, the “why” behind the gains remains somewhat opaque.
2. Moreover, the approach introduces additional training overhead through diffusion sampling and warm-up phases, but the paper does not quantify these costs or discuss trade-offs between sample diversity and efficiency. Finally, while the method is positioned as general, all evaluations are within classification tasks; its potential limitations on forecasting, regression, or multimodal fusion remain untested, leaving open questions about scalability beyond the current scope.

**Questions:**

1. To what extent does the diffusion prior capture epistemic uncertainty about the teacher’s manifold, as opposed to merely adding stochastic regularization? Could visualizing the learned diffusion trajectories help clarify this distinction?
2. The GDPD assumes the teacher’s feature space forms a coherent generative manifold. How sensitive is the approach to teacher architectures where intermediate representations are poorly structured?

---

> ### Author Response · Authors · 2025-12-03
> **Author Response to Reviewer  VsYh**
>
> We sincerely thank the reviewer for their thoughtful evaluation and for recognizing the novelty of framing the teacher’s representation as a generative prior, as well as the breadth and systematic design of our experiments. We address the reviewer’s remaining concerns below.
>
>
> >**Q1.** To what extent does the diffusion prior capture epistemic uncertainty about the teacher’s manifold, as opposed to merely adding stochastic regularization?
>
> **A1.** Thank you for raising this interesting point. A rigorous uncertainty-quantification analysis lies beyond the scope of our main contribution, which focuses on knowledge distillation under partial observability. However, in response to the reviewer’s concern, we include a focused experiment to provide additional insight for interested readers.
>
> Epistemic uncertainty reflects model unfamiliarity and manifests in feature space as weakly supported or unstable representations. Prior work quantifies this by measuring representation support [1,2], since unfamiliar samples typically lie in sparsely populated regions. Following this, we measure local support using the k-nearest-neighbour (k = 10) distance as a simple proxy for epistemic uncertainty.
> First, we compute the local support for each test sample in the teacher feature space, using full-length teacher representations. Then, using the diffusion prior trained with partialness
> $e=0.5L$, we generate posterior reconstructions for each test sample conditioned on the corresponding short-context student feature. For these reconstructed features, we again compute local support using the same kNN procedure.
>
> Finally, we measure the correlation between local support measures of teacher features and diffusion-reconstructed features. If the diffusion prior merely added stochastic noise, these two quantities would be largely uncorrelated. Instead, we observe a moderate monotonic relationship (Spearman $r = 0.54$, $p < 10^{-300}$) and a moderate linear association (Pearson $r = 0.44$, $p < 10^{-300}$). This indicates that the diffusion prior preserves the teacher’s epistemic structure: samples that the teacher places in low-support (high-uncertainty) regions also tend to receive low support in the reconstruction space.
>
>
> [1] Van Amersfoort, J., Smith, L., Teh, Y. W., & Gal, Y. (2020, November). Uncertainty estimation using a single deep deterministic neural network. In International conference on machine learning (pp. 9690-9700). PMLR.
>
> [2] Kumar, D., Poggi, P., Tayebati, S., Naik, D., Ahuja, N., & Trivedi, A. R. (2025). Calibrated Decomposition of Aleatoric and Epistemic Uncertainty in Deep Features for Inference-Time Adaptation. arXiv preprint arXiv:2511.12389.
>
>
> >**W1.**  The “why” behind the gains.
>
> **A2.** Sections 3.3 and 3.4 explain how the distributional teacher knowledge benefits GDPD by generating teacher signals that (1) are dynamic and progressive with
> respect to the student’s current capability, (2) provide stochastic diversity of the same features, and (3) faithful knowledge through collective aggregation. These properties together explain why GDPD provides stronger supervision than conventional KD.
>
> We empirically validate that each of these properties drives the performance gains through: (1) ablations replacing GDPD with deterministic teacher supervision (Section 4.2), (2) ablations manipulating the diversity of GDPD (Sections 4.2 and A.3.1), and (3) confirmation of faithful knowledge transfer via fidelity analysis (Section 4.1, Figure 2), respectively.
>
>
> > **W2.** The paper does not quantify computational costs or discuss trade-offs between sample diversity and efficiency.
>
> **A3.** Thank you for this valuable feedback. We kindly refer the reviewer to the “Author response on the computational cost of GDPD” comment for the detailed response. Incorporating the reviewer’s suggestions into the revised manuscript, we have added a comprehensive computational cost evaluation in Section A.3.2.
> We clarify that the additional overhead introduced by GDPD is modest, incurred only during training, and does not affect inference. We kindly refer the reviewer to Section A.3.2 for a detailed analysis of the computational cost of each phase of GDPD’s training, the comparison with baseline methods, and cost–performance trade-offs from diffusion controls (including sample diversity).
> For additional details on the performance impact of diffusion controls and other hyperparameters, we kindly refer the reviewer to Sections 4.2 and A.3.1.

---

> ### Author Response · Authors · 2025-12-03
> **Author Response to Reviewer VsYh**
>
> >**W2.** GDPD’s potential limitations on forecasting, regression, or multimodal fusion remain untested, leaving open questions beyond the current scope.
>
> **A4.** Our primary goal in this work is to establish GDPD as a principled distillation framework for partial time-series classification, and we validate it extensively across UCR benchmarks, UEA benchmarks, and a real-world PhysioNet case study, together with practical guidance on diffusion controls and key hyperparameter selection.
> The evaluated implementation supervises posterior reconstructions using a classification loss and is therefore instantiated for classification. However, modeling teacher knowledge as a generative prior constitutes a novel and broadly applicable perspective on knowledge distillation: by supervising the posterior reconstructions with the relevant task loss, GDPD can in principle be adapted to forecasting, regression, or multimodal encoder settings. Evaluating these additional task families would require substantial task-specific preprocessing and evaluation pipelines—amounting to separate lines of investigation that fall beyond the scope of this paper. In light of the reviewer’s concerns, we have updated the “Limitations and Future Work” section (Appendix A.4) to discuss these extensions.
>
> >**Q2.** How sensitive is the approach to teacher architectures where intermediate representations are poorly structured?
>
> **A5.** Thank you for this valuable feedback. We have revised the paper to include experiments addressing this question in Section A.3.4 (*Robustness Under Weak-Teacher Supervision*). To assess robustness when the teacher provides poorly structured feature spaces, we construct a sequence of increasingly degraded teachers by reducing the amount of training data and injecting label noise. As teacher quality degrades, GDPD maintains stable performance compared to the baselines, demonstrating robustness to poorly structured teacher supervision. We kindly refer the reviewer to Section A.3.4 for a detailed discussion.
>
> We sincerely appreciate the reviewer’s thoughtful and constructive feedback, which has greatly strengthened our work. We hope that our detailed revisions and clarifications satisfactorily address all concerns, and we would be grateful if the reviewer could consider this progress in their final evaluation

---

### Official Review · Reviewer_yLkg · 2025-10-31

**Soundness:** 2
**Presentation:** 2
**Contribution:** 2
**Rating:** 4
**Confidence:** 3

**Summary:**

The paper tackles classification from partial time-series prefixes by distilling knowledge from a teacher trained on full sequences.
Extensive experiments across earliness settings, datasets, and architectures demonstrate GDPD’s effectiveness for full-to-partial distillation

**Strengths:**

Generally clear and readable; figures and tables are informative; Codes are publicly available.

**Weaknesses:**

1. Training a diffusion model may additionally introduce significant computation and memory
2. compared baselines (KD and FitNet) are significantly out-of-dated
3. the paper lacks a formalization of the posterior being sampled

**Questions:**

see weakness

---

> ### Author Response · Authors · 2025-11-25
> **Author Response to Reviewer  yLkg**
>
> We sincerely thank the reviewer for the helpful comments and constructive suggestions, which have improved the clarity and presentation of our work. To address the reviewer’s concerns, we clarify each of their points below.
>
> > **W1.**  Training a diffusion model may additionally introduce computation and memory.
>
> **A1.** Thank you for this valuable feedback.  We kindly refer the reviewer to the “Author response on the computational cost of GDPD” comment for the detailed response. Incorporating the reviewer’s suggestions into the revised manuscript, we have added a comprehensive computational cost evaluation in Section A.3.2.
> We clarify that the additional overhead introduced by GDPD is modest, incurred only during training, and does not affect inference. GDPD adds only a small training-time increase over logit-based KD and remains comparable to established feature-distillation methods such as RKD and VID, while achieving substantially higher performance. Its memory footprint is similarly low: close to FitNet and VID, slightly above logit-based KD, and far below memory-intensive methods like RKD. We kindly refer the reviewer to Section A.3.2 for a detailed analysis of the computational cost of each phase of GDPD’s training, the comparison with baseline methods, and the cost analysis of diffusion controls.

---

> ### Author Response · Authors · 2025-11-25
> **Author Response to Reviewer yLkg**
>
> > **W3.** the paper lacks a formalization of the posterior being sampled
>
> **A2.** We thank the reviewer for the comment. In GDPD, we model the conditional distribution $p_{\Phi}(z_{long} \mid z_{short})$ which represents the posterior over full-context features given the student's short-context feature. This posterior searches over the space of $z_{long}$ for the full-context representation that best explains the observed $z_{short}$, regarding $z_{short}$ as a degraded or partial observation of $z_{long}$.
>
> To approximate this conditional sampler, GDPD modifies the reverse process of the diffusion prior learned on teacher features $p_{\phi}(z_{long})$ so that $z_{short}$ influences the generative trajectory. Specifically, the reverse process is initialized at $z_{long,T}$ with a transformation of $z_{short}$, enabling the sampler to explore the teacher-feature manifold while remaining anchored to the initialization and converging toward representations consistent with the information present in $z_{short}$.
> The resulting sampling procedure therefore constitutes a valid implicit posterior over $z_{long}$ conditioned on ${z}_{short}$. We present this formalization in Equations (1)–(6), and the underlying logic is further detailed in the corresponding text. We appreciate the reviewer’s feedback and would be happy to further clarify any remaining aspects.
>
>
> > **W2.** compared baselines (KD and FitNet) are out-of-dated
>
> **A3.**  We compare GDPD against widely adopted distillation methods, including RKD (2019), Attention (2016), DKD (2022), DT2W (2023), and VID (2019), along with FitNet and Base-KD—in Section 4.1 (*GDPD Outperforms Existing KD Variants*).
> In response to the reviewer’s request to incorporate more recent distillation methods, we additionally evaluated three newer methods: TTM [1] (2024), PKT [2] (2020), and TeKAP [3] (2025). The updated results table is provided below.  Base-KD and FitNet were prioritised as primary baselines because they represent the standard logit-based and feature-based distillation frameworks commonly used to assess the effectiveness of newly proposed distillation objectives, and thus serve as key reference points for evaluating GDPD.
>
>
>
> **Table 2: Comparison of distillation methods at earliness $e = 0.5L$ on 12 UCR datasets. Test-fidelity is reported as average top-1 agreement. Rows marked with $\downarrow$ indicate lower-is-better.**
>
> | Method            |  Base | Base-KD |  Fits |  VID  |  DKD  |  Att. |  DT2W |  RKD  | RKD-Ang. |  PKT  | TeKAP |  TTM  |    GDPD   |
> | :---------------- | :---: | :-----: | :---: | :---: | :---: | :---: | :---: | :---: | :------: | :---: | :---: | :---: | :-------: |
> | **Avg. AUC-PRC**  |  71.6 |  78.14  | 77.28 | 77.19 | 74.29 | 75.47 | 55.32 | 79.89 |   79.26  | 76.86 | 76.54 | 78.05 | **84.64** |
> | **Test-Fidelity** | 66.75 |  72.13  | 69.39 | 71.32 | 69.36 | 69.20 | 52.59 | 74.34 |   71.22  | 71.05 | 69.53 | 70.24 | **77.58** |
> | **Avg. Rank (↓)** |  9.67 |   5.83  |  7.58 |  6.33 |  8.25 |  7.25 | 11.42 |  6.33 |   6.42   |  6.67 |  6.75 |  6.17 |  **2.25** |
> | **Num. Top-1**    |   0   |    2    |   0   |   0   |   0   |   0   |   0   |   2   |     2    |   0   |   1   |   0   |   **5**   |
> | **Num. Top-3**    |   0   |    5    |   2   |   1   |   1   |   2   |   0   |   4   |     4    |   3   |   2   |   2   |   **10**  |
> | **Wins/Draws**    |   12  |    10   |   12  |   11  |   11  |   11  |   12  |   9   |     9    |   10  |   11  |   11  |     —     |
> | **Losses**        |   0   |    1    |   0   |   1   |   1   |   1   |   0   |   3   |     3    |   2   |   1   |   1   |     —     |
>
>
> [1] Zheng, Kaixiang, and En-Hui Yang. “Knowledge distillation based on transformed teacher matching.” arXiv preprint arXiv:2402.11148 (2024). (ICLR 2024)
>
> [2] Passalis, Nikolaos, Maria Tzelepi, and Anastasios Tefas. “Probabilistic knowledge transfer for lightweight deep representation learning.” IEEE Transactions on Neural Networks and Learning Systems 32, no. 5 (2020): 2030–2039.
>
> [3] Hossain, Md Imtiaz, Sharmen Akhter, Choong Seon Hong, and Eui-Nam Huh. “Single teacher, multiple perspectives: Teacher knowledge augmentation for enhanced knowledge distillation.” ICLR 2025.
>
> We sincerely appreciate the reviewer’s thoughtful and constructive feedback, which has greatly strengthened our work. We hope that our detailed revisions and clarifications satisfactorily address all concerns, and we would be grateful if the reviewer could consider this progress in their final evaluation.

---

### Official Review · Reviewer_y3wg · 2025-11-01

**Soundness:** 3
**Presentation:** 3
**Contribution:** 2
**Rating:** 6
**Confidence:** 3

**Summary:**

This paper proposes a new method to train time series classifiers that operate on partial sequence prefixes to achieve similar generalization performance to those trained on full sequences. The authors argue that the generalization gap in this case is due to a representation gap, as the full context features from teacher models cannot be directly distilled into the partial context features from student models. To solve this, a diffusion model is trained to obtain a generative prior over full context features, which can then be used to posterior-sample completed features given the partial features from the student model. The student model is then optimized to produce partial features such that its corresponding posterior full feature samples have high classification performance.

**Strengths:**

1. This paper introduces an interesting formulation for the partial time series prefix classification problem, which identifies the representation gap as a key reason why conventional distillation methods do not work as expected.

2. The proposed method is novel, which leverages powerful diffusion priors to obtain posterior full features given partial features for the downstream classification task. The learning objective is effective since it directly optimizes to student model to achieve highly discriminative performance given the posterior full feature samples.

3. The authors provide some reasonable justifications for why the proposed method works, which is much appreciated.

4. Extensive experiments with ablation studies are performed the validate the proposed method's empirical performance. However, I cannot evaluate the experimental results, since I'm not familiar with time series classification benchmarks and baselines.

**Weaknesses:**

1. My main concern is the computational complexity of the proposed method in all training stages. First, in the pre-training stage, it involves pre-training (i) a teacher classifier on full sequences; (ii) a diffusion model on the full features extracted from one of the middle layers of the teacher classifier. Then, when training the student model, each training step requires sampling from the posterior distribution, which involves simulating the reverse diffusion process of the diffusion model for multiple steps. This is significantly more costly than traditional distillation methods.

2. It is unclear why the noise-fusing approach is used for the conditioning mechanism for the diffusion model, given that classifier-guided/free guidance is more principled and widely used in many applications.

3. Limitations of the proposed method is not discussed in the paper.

**Questions:**

1. Could the authors comment on and provide some analysis of the trade-off between model performance and computational cost for the proposed method and other conventional distillation methods?

2. Could the authors clarify why the noise-fusing approach is used for diffusion model conditioning? How does it compared to classifier-free/guided guidance in this setting?

3. Could the authors discuss limitations of the proposed method in the paper?

4. Does z_long and z_short have the same dimensionality? From the paper, it is unclear whether the posterior sampling procedure is an impainting process that fills in the missing postfix in z_short or a transformation that modifies the entirety of z_short into z_long.

---

> ### Author Response · Authors · 2025-11-24
> **Response to Reviewer y3wg**
>
> We sincerely thank the reviewer for their thoughtful evaluation and for finding our formulation interesting, as well as for noting the effectiveness of our diffusion-prior design and the strength of our empirical and ablation results. To address the reviewer’s remaining concerns, we clarify each of their points below.
>
> > **Q1 & W1.** Could the authors comment on and provide some analysis of the trade-off between model performance and computational cost for the proposed method and other conventional distillation methods?
>
> **A1** Thank you for this valuable feedback. We kindly refer the reviewer to the “Author response on the computational cost of GDPD” comment for the detailed response. Incorporating the reviewer’s suggestions into the revised manuscript, we have added a comprehensive computational cost evaluation in Section A.3.2.
>
>  We clarify that the additional overhead introduced by GDPD is modest, incurred only during training, and does not affect inference. GDPD adds only a small training-time increase over logit-based KD and remains comparable to established feature-distillation methods such as RKD and VID, while achieving substantially higher performance. Its memory footprint is similarly low: close to FitNet and VID, slightly above logit-based KD, and far below memory-intensive methods like RKD. We kindly refer the reviewer to Section A.3.2 for a detailed analysis of the computational cost of each phase of GDPD’s training, the comparison with baseline methods, and the cost analysis of diffusion controls.

---

> ### Author Response · Authors · 2025-11-24
> **Response to Reviewer y3wg**
>
> > **Q2 & W2.** Could the authors clarify why the noise-fusing approach is used for diffusion model conditioning? How does it compared to classifier-free/guided guidance in this setting?
>
> **A2.** We thank the reviewer for raising this question. Classifier guidance and classifier-free guidance are typically used in guided sampling where the objective is to steer the diffusion model toward a desired semantic or class-conditioned output. In contrast, GDPD operates in an inverse diffusion setting, where the guidance is provided by the measurement or partial observation. The conditioning mechanism in inverse diffusion is fundamentally different from classifier- or classifier-free guidance used in guided generation. Below, we justify our use of simple initialization-based conditioning in GDPD by outlining why other common conditioning strategies in the inverse-diffusion are unsuitable for our setting.
>
> **(1) Conditional denoisers.**
> A common approach is to feed the measurement ($z_{\text{short}}$) into the denoiser during both training and sampling, allowing the denoiser to learn how to influence the output $z_{\text{long}}$. This resembles classifier-free guidance in guided generation. However, this is incompatible with GDPD since conditioning signals are non-stationary. During training, the student keeps learning, and its $z_{\text{short}}$ features change continuously. A conditional denoiser trained on earlier student states quickly becomes invalid, because it no longer reflects the student’s updated representation. Keeping it aligned would require retraining the denoiser at every student update. For this reason, GDPD cannot rely on conditional denoisers.
>
> **(2) Projection or data-consistency updates.**
> These methods apply an unconditional denoiser and then project the output onto the set of measurements to enforce consistency with the observation, typically assuming a known, fixed, and linear degradation model. In GDPD, the “degradation’’ from $z_{\text{long}}$ to $z_{\text{short}}$ is neither known, fixed, nor linear. The student features $z_{\text{short}}$ evolve throughout training, making any strict projection-based constraint incompatible with their non-stationary nature.
>
> **(3) Bayesian likelihood-gradient conditioning.**
> These methods use an unconditional denoiser and modify the score at each diffusion step with the gradient of the likelihood—i.e., how likely the current sample would produce the observed measurement. Because this likelihood is intractable, inverse-diffusion methods approximate it using assumptions tied to their specific degradation setting (linear, nonlinear, noisy, etc.). This requires explicit knowledge of the forward degradation and introduces significant computational overhead. In GDPD, however, the degradation from $z_{\text{long}}$ to $z_{\text{short}}$ is unknown, dynamic, and not tied to any fixed physical process. Because the GDPD loss already supervises the outputs and the conditioning signal itself evolves during training, we treat this mapping as general and do not enforce strict measurement consistency at every diffusion step. Imposing such exact reconstruction constraints would be both unnecessary and computationally costly.
>
> **(4) Initialization-based conditioning.**
> We adopt initialization-based conditioning because it accommodates the evolving nature of the student features (with the appropriate noise level) without requiring any retraining or modification of the conditioning mechanism. As the student updates, its $z_{\text{short}}$ features directly influence the diffusion trajectory, ensuring that the conditioning always reflects the student’s current knowledge. Although this approach does not enforce strict measurement consistency, this is acceptable: the GDPD loss already supervises the reconstructed features, and the conditioning signal only needs to steer the generation process rather than impose exact reconstruction constraints.
>
> > **Q3 & W3.** Could the authors discuss limitations of the proposed method in the paper?
>
> **A3.** Thank you for the suggestion. We have added a *Limitations and Future Work* section in Appendix A.4, outlining the limitations of GDPD and noting that modeling teacher knowledge as a generative prior opens up a broader family of distillation objectives, of which the current work instantiates only one, leaving alternative formulations for future work.

---

> ### Author Response · Authors · 2025-11-24
> **Response to Reviewer y3wg**
>
> > **Q4.** Does z-long and z-short have the same dimensionality? From the paper, it is unclear whether the posterior sampling procedure is an impainting process that fills in the missing postfix in z-short or a transformation that modifies the entirety of z-short into z-long.
>
>  **A4.** In GDPD, $z_{\text{short}}$ and $z_{\text{long}}$ have the same dimensionality.
> When student and teacher feature dimensions differ (e.g., under model compression), we apply lightweight learnable projections (convolutions) to map $z_{\text{short}}$ into the teacher dimension and to project the reconstructed features back into the student dimension.
> The posterior sampling is not an inpainting process. Instead, it treats $z_{\text{short}}$ as a partial or weaker representation of the full-context feature and performs a global transformation. The meaning of the $W$ symbols in Figure 1(a) has been clarified, and additional implementation details have been added in Appendix A.2.
>
> We sincerely appreciate the reviewer’s thoughtful and constructive feedback, which has greatly strengthened our work. We hope that our detailed revisions and clarifications satisfactorily address all concerns, and we would be grateful if the reviewer could consider this progress in their final evaluation.

---

### Official Review · Reviewer_uuK6 · 2025-11-02

**Soundness:** 3
**Presentation:** 3
**Contribution:** 3
**Rating:** 6
**Confidence:** 3

**Summary:**

This paper studies classification from partial time-series prefixes and proposes GDPD, a distillation framework that learns a diffusion prior over teacher features and uses posterior samples—conditioned on the student’s short-context features—to supervise the student. The aim is to provide progressive, diverse, and distributional targets rather than single-point matches. Across UCR/UEA benchmarks and a PhysioNet case study, GDPD improves accuracy and teacher–student fidelity under multiple earliness levels, channel-wise partialness, compression, and self-distillation.

**Strengths:**

- The work focuses on full-context → partial-context distillation and explains why direct feature/logit matching can overwhelm the student under representation mismatch.
- Treating teacher representations as a diffusion-learned prior and supervising with distributional, progressive targets is a substantive shift from standard KD.
- Consistent gains over Base, Fits, and other KD baselines across earliness settings; better teacher–student agreement; robustness to channel-wise partialness; and utility for compression and self-distillation.
- Warm-up scheduling, loss-weight trade-offs between task and GDPD, and the effect of sampling multiplicity J provide insight into what drives improvements.
- The PhysioNet ICU study suggests the approach remains effective under heterogeneous conditions (partial channels, imbalance, cross-task).

**Weaknesses:**

- Training adds a diffusion prior and posterior sampling within student optimization. Please report wall-clock and step-level overhead versus Fits/Logit-KD (e.g., epochs × diffusion steps, training/inference cost).
- Performance may depend on which teacher layer is modeled and which student features are aligned. Clear guidance, cross-layer experiments, or robustness checks would help practitioners.
- Please discuss applicability beyond time series (e.g., images or text under partial inputs) and clearly state assumptions and limitations to contextualize the results and guide follow-up work.

**Questions:**

- How is $z_{\text{long-hint}}$ defined for a given example—what subset of the teacher feature manifold counts as useful hints of $z^*_{\text{long-ideal}}$? If $z_{\text{long-hint}}\sim p(z_{\text{long}})$ is just a generic draw, how does it differ from standard teacher features, and why doesn’t this construct appear later in the method/experiments? Please clarify the relationship between the posterior reconstruction $\hat z_{\text{long}}$ and the hints, or state that $z_{\text{long-hint}}$ is expository only.
- Which teacher/student layers define $z_{\text{long}}$ and $z_{\text{short}}$ across architectures? Have you tried cross-layer modeling (multiple blocks) or multi-scale priors?
- How much training time does GDPD add on typical settings, and how sensitive are results to diffusion steps/samples and key hyperparameters? Any guidance on trade-offs between speed and accuracy?

---

> ### Author Response · Authors · 2025-11-30
> **Author Response to Reviewer uuK6.**
>
> We greatly appreciate the reviewer’s thoughtful evaluation and their recognition of the methodological contribution, noting that our use of progressive, distributional targets represents a substantive shift from standard KD.
> To address the reviewer’s remaining concerns, we clarify each of their points below.
>
> > **Q1.**  How is $z_{\text{long-hint}}$ defined for a given example—what subset of the teacher feature manifold counts as useful hints of $z^*_{\text{long-ideal}}$? If $z_{\text{long-hint}} \sim p(z_{long})$ is just a generic draw, how does it differ from standard teacher features, and why doesn’t this construct appear later in the method/experiments? Please clarify the relationship between the posterior reconstruction $\hat z_{long}$
>  and the hints, or state that $z_{\text{long-hint}}$ is expository only.
>
>
> **A1.** We thank the reviewer for the question. We use the term $z_{\text{long-hint}}$ expositorily to denote the subset of teacher features that contains the correct long-context knowledge needed to classify a given sample, and is characterized by the predictive property—i.e., the ability to assign the correct label while remaining consistent with the evidence already captured in the short-context features. Each $z_{\text{long-hint}}$ is assumed to be a close approximation (or valid completion) of the ideal long-context feature $z*_{\text{long-ideal}}$, serving as its proxy.
> If the student has learned short-context features that induce minimal degradation relative to $z*_{\text{long-ideal}}$, then student features provide the right conditioning for the diffusion model to recover a representation close to $z^*_{\text{long-ideal}}$—i.e., a “hint’’ feature. Therefore, $z_{\text{long-hint}}$ also characterizes the the posterior reconstructions that the student would produce under the desired optimal parameters.
>
> The relationship between $\hat z_{long}$ and the “hints’’ is given explicitly in Equation 3: the posterior reconstructions obtained under optimal student parameters ($\hat z_{long}; \theta *$) correspond directly to the hint features.
>
>
> > **Q2.** Which teacher/student layers define $z_\text{{long}}$ and $z_\text{{short}}$ across architectures? Have you tried cross-layer modeling (multiple blocks) or multi-scale priors?
> **W2.** Clear guidance, cross-layer experiments, or robustness checks would help practitioners.
>
> **A2.** We thank the reviewer for this valuable feedback. In response, we conducted additional experiments and revised the manuscript to include Section A.3.3, which provides clearer guidance on GDPD’s layer selection. We present a comprehensive cross-layer analysis evaluating all teacher–student layer combinations, along with a multi-layer prior trained using multiple teacher layers.
> Our results show that final-layer distillation gives the strongest performance, while distillation from shallow layers performs slightly lower yet remains close. Incorporating multi-layer priors is feasible and results in stable performance, but does not outperform final-layer distillation, suggesting deep teacher features alone are the most effective. We additionally report layer-wise averages and rank calculations for both teacher and student layers, further confirming that the final layer is the most effective choice.
> For all experiments in the main paper, $z_{\text{long}}$ and $z_{\text{short}}$ are extracted from the final feature layer of the teacher and the student.
>
> >**W3.** Please discuss applicability beyond time series and clearly state assumptions and limitations to guide follow-up work.
>
> **A3.** Thank you for this valuable feedback. Considering the reviewer’s suggestion, we have revised the *Limitations and Future Work* section (Section A.4) to include a discussion on “Applicability Beyond Time Series”, where we outline the key considerations for adapting GDPD to other domains.

---

> ### Author Response · Authors · 2025-12-03
> **Author Response to Reviewer uuK6.**
>
> > **Q3.**  How much training time does GDPD add on typical settings, and how sensitive are results to diffusion steps/samples and key hyperparameters? Any guidance on trade-offs between speed and accuracy?
> **W1.** Please report wall-clock and step-level overhead versus Fits/Logit-KD (e.g., epochs × diffusion steps, training/inference cost
>
> **A4.** Thank you for this valuable feedback. We kindly refer the reviewer to the “Author response on the computational cost of GDPD” comment for the detailed response.
> Incorporating the reviewer’s suggestions into the revised manuscript, we have added a comprehensive computational cost evaluation in Section A.3.2.
>
> We clarify that the additional overhead introduced by GDPD is modest, incurred only during training, and does not affect inference. In summary, GDPD adds only a small training-time increase (0.24 s/epoch) over logit-based KD and remains comparable to established feature-distillation methods such as RKD and VID, while achieving substantially higher performance. Its memory footprint is similarly low: close to FitNet and VID, slightly above logit-based KD, and far below memory-intensive methods like RKD.
> Section A.3.2 reports both wall-clock and step-level overhead, detailing the computational cost of each phase of GDPD’s training, the comparison with baseline methods (Fits and Logit-KD), and the cost analysis of diffusion-control settings (diffusion steps and posterior samples). For additional details on the performance impact of diffusion controls and other hyperparameters, we kindly refer the reviewer to Sections 4.2 and A.3.1.
>
> We sincerely appreciate the reviewer’s thoughtful and constructive feedback, which has greatly strengthened our work. We hope that our detailed revisions and clarifications satisfactorily address all concerns, and we would be grateful if the reviewer could consider this progress in their final evaluation.

---

### Author Response · Authors · 2025-11-24
**Author response to reviewer comments on the computational cost of GDPD  (2/2).**

(Continued)

We discuss the computational cost of each phase of GDPD below.

**Teacher training.**  Training the teacher is identical to any standard KD pipeline and introduces no additional cost in GDPD.

**Diffusion-prior training.**
Trained in feature space, where the dimensionality is far lower than in the input domain, GDPD’s diffusion training becomes significantly cheaper computationally. The diffusion prior is lightweight (only 0.206M trainable parameters, including the noise adapter) and is trained during the student’s warm-up phase.
The warm-up stage costs 0.81 s/epoch, compared to 0.61 s/epoch for the Base student. With a warm-up duration of 300 epochs, this adds only $\sim$1 minute of extra training in the evaluated setting. In practice, diffusion-prior training is comparable to training one additional Base classifier and remains far cheaper than ensemble-teacher distillation, while still providing the benefit of knowledge diversity.

**Diffusion-guided training.**
During GDPD’s main distillation phase, posterior sampling uses only 5 NFEs, and all sampling is performed in feature space, making each reverse-diffusion step extremely cheap. The diffusion-guided stage costs 0.89 s/epoch, compared to 0.61 s/epoch for the Base student, an overhead of only $\sim$0.28 s per epoch. Over 300 epochs, this amounts to approximately 1.2 minutes of additional training time on the evaluated dataset.

**Overall training cost.**
In the evaluated setting, training the Base and Base-KD requires 0.10 h, whereas GDPD (warm-up + guided phase) requires 0.14 h, adding only $\sim$2.4 minutes of extra training. The overall training cost of GDPD is comparable to widely adopted feature-distillation methods such as RKD and VID, while delivering substantially higher performance (Table 2).
GDPD also maintains a low memory footprint of 0.24-0.25 GB, similar to Fits and VID, only slightly above logits-based KD (0.17~GB), and far below memory-intensive methods such as RKD (1.03 GB). Despite incorporating a diffusion prior, GDPD introduces minimal memory overhead.
This modest training overhead is well justified by the consistent and significant performance improvements over conventional KD baselines and the Base classifier (Table 2).

**Inference cost.**  Inference cost is unchanged: the GDPD achieves 0.02~ms/sample, identical to the Base. All additional computation occurs only during training, while the deployed model remains as efficient as the Base classifier.

**Training cost vs. inference steps.**
Table 14 reports how the computational cost varies with the number of inference steps (equivalently, the NFEs in our implementation).
Increasing inference steps slightly raises per-epoch time and memory. From 1 to 5 steps (our default), epoch time increases from 0.77 s to 0.85 s and memory from 0.24 GB to 0.25 GB. Even at 10 steps, memory remains at 0.25 GB and total training stays below 0.16 h. We adopt 5 steps as the best cost-performance trade-off.


**Training cost vs. posterior samples.**
Table 15 summarizes how training cost scales with the number of posterior samples. The cost grows roughly linearly, as each sample requires an additional draw. Epoch time rises from 0.85 s (1 sample) to 1.44 s (5 samples), and total training from 0.14 h to 0.24 h, while GPU memory remains fixed at 0.25 GB. We adopt a single sample as the best cost-performance trade-off.

Both ablations show that GDPD’s diffusion controls offer a flexible cost-performance trade-off with minimal memory overhead and no effect on inference speed.

---

### Author Response · Authors · 2025-11-24
**Author response to reviewer comments on the computational cost of GDPD (1/2).**

**Q.**  Reviewer questions regarding the computational cost evaluation of GDPD

**A.** We clarify that the computational overhead of GDPD is modest, incurred only during training, and does not affect inference. To quantify this, we evaluated the training cost on the StarLightCurves dataset for the $\text{LSTM3-100} \rightarrow \text{LSTM3-100}$. Table 13 summarizes the results obtained using a single RTX~A6000 / 3090–class GPU.

**Table 13. Training, memory, and inference cost comparison across distillation methods.**

| **Method** | **Student**  **Params (M)** | **Additional**  **Params (M)** | **Total**  **Train (h)** | **Epoch**  **Time (s)** | **Step**  **Time (ms)** | **GPU Mem.** **(GB)** | **Inference**  **(ms)** |
|-----------|----------------------------------|--------------------------------------|------------------------------|------------------------------|------------------------------|------------------------------|------------------------------|
| Base               | 0.20 | 0    | 0.10 | 0.61 | 46.71 | 0.17 | 0.02 |
| Base KD            | 0.20 | 0    | 0.10 | 0.61 | 46.88 | 0.17 | 0.02 |
| Fits               | 0.20 | 0.01 | 0.10 | 0.62 | 48.34 | 0.22 | 0.02 |
| VID                | 0.20 | 0.03 | 0.11 | 0.68 | 52.00 | 0.25 | 0.02 |
| DKD                | 0.20 | 0    | 0.10 | 0.62 | 47.98 | 0.17 | 0.02 |
| Attention          | 0.20 | 0    | 0.11 | 0.64 | 49.00 | 0.17 | 0.02 |
| RKD                | 0.20 | 0    | 0.13 | 0.80 | 61.61 | 1.03 | 0.02 |
| GDPD               | 0.20 | 0.21 | 0.14 | 0.85 | 65.14 | 0.25 | 0.02 |
| GDPD warm-up       | 0.20 | 0.21 | 0.07 | 0.81 | 62.46 | 0.24 | 0.02 |
| GDPD guided phase  | 0.20 | 0.21 | 0.08 | 0.89 | 69.21 | 0.25 | 0.02 |


**Table 14. Effect of inference diffusion steps on training cost, memory, and inference time.**

| Inference Steps | Total Train (h) | Epoch Time (s) | Step Time (ms) | GPU Mem (GB) | Inference (ms) |
|-----------------|-----------------|----------------|----------------|--------------|----------------|
| 0               | 0.10            | 0.61           | 46.71          | 0.17         | 0.02           |
| 1               | 0.13            | 0.77           | 59.03          | 0.24         | 0.02           |
| 2               | 0.13            | 0.80           | 61.86          | 0.24         | 0.02           |
| 3               | 0.13            | 0.81           | 62.45          | 0.25         | 0.02           |
| 5               | 0.14            | 0.85           | 65.14          | 0.25         | 0.02           |
| 10              | 0.16            | 0.95           | 72.79          | 0.25         | 0.02           |


**Table 15. Effect of the number of posterior samples on training cost, memory, and inference time.**

| Posterior Samples (J) | Total Train (h) | Epoch Time (s) | Step Time (ms) | GPU Mem (GB) | Inference (ms) |
|-------------------|-----------------|----------------|----------------|--------------|----------------|
| 0                 | 0.10            | 0.61           | 46.71          | 0.17         | 0.02           |
| 1                 | 0.14            | 0.85           | 65.14          | 0.25         | 0.02           |
| 2                 | 0.17            | 0.99           | 76.05          | 0.25         | 0.02           |
| 3                 | 0.19            | 1.11           | 85.99          | 0.25         | 0.02           |
| 4                 | 0.22            | 1.30           | 100.17         | 0.25         | 0.02           |
| 5                 | 0.24            | 1.44           | 111.07         | 0.25         | 0.02           |

---

### Author Response · Authors · 2025-12-03
**Author Summary for the Area Chair**

Due to the recent issue in OpenReview and the subsequent system freeze, none of our reviewers were able to participate in the discussion phase. As a result, they did not have the opportunity to engage with or respond to our clarifications. Our detailed revisions and clarifications address all concerns raised in the reviews, and we hope they assist you during the meta-review process. We sincerely appreciate your time and consideration.

---

### Meta-Review · Program_Chairs · 2026-01-04

**Summary:**

This paper proposes Generative Diffusion Prior Distillation (GDPD), a novel knowledge distillation (KD) framework designed to transfer generalization capability from a teacher trained on full-length time series to a student operating on partial (prefix) sequences. The core idea is to model teacher knowledge as a generative diffusion prior, treating student features as degraded observations of ideal full-context features. Through diffusion-based posterior sampling, GDPD provides the student with diverse, progressive, and collective teacher signals, rather than a single static target. Experiments across multiple earliness levels, architectures, and datasets show that GDPD outperforms existing KD methods in terms of classification performance (AUC-PRC), fidelity to the teacher, and robustness to partialness in both time and channel dimensions.

**Reviewer Scores:**

No

---

### Decision · Program_Chairs · 2026-01-26

Accept (Poster)